# 3D collagen architecture induces a conserved migratory and transcriptional response linked to vasculogenic mimicry

D.O. Velez[1], B. Tsui [2], T. Goshia[1], C.L. Chute[1], A. Han[1], H. Carter [3,4] & S.I. Fraley[1,4]

The topographical organization of collagen within the tumor microenvironment has been implicated in modulating cancer cell migration and independently predicts progression to metastasis. Here, we show that collagen matrices with small pores and short fibers, but not Matrigel, trigger a conserved transcriptional response and subsequent motility switch in cancer cells resulting in the formation of multicellular network structures. The response is not mediated by hypoxia, matrix stiffness, or bulk matrix density, but rather by matrix architecture-induced β1-integrin upregulation. The transcriptional module associated with network formation is enriched for migration and vasculogenesis-associated genes that predict survival in patient data across nine distinct tumor types. Evidence of this gene module at the protein level is found in patient tumor slices displaying a vasculogenic mimicry (VM) phenotype. Our findings link a collagen-induced migration program to VM and suggest that this process may be broadly relevant to metastatic progression in solid human cancers.

[1] Department of Bioengineering, University of California, San Diego, La Jolla, CA 92093, USA. [2] Bioinformatics and Systems Biology Program, University of California, San Diego, La Jolla, CA 92093, USA. [3] Department of Medicine, University of California, San Diego, La Jolla, CA 92093, USA. [4] Moores Cancer Center, University of California, San Diego, La Jolla, CA 92093, USA. Correspondence and requests for materials should be addressed to S.I.F. (email: sifraley@ucsd.edu)

An initial step in cancer metastasis is the migration of tumor cells through the extracellular matrix (ECM) and into the lymphatic or vascular systems[1]. Several features of the tumor ECM have been associated with progression to metastasis. In particular, regions of dense collagen are co-localized with aggressive tumor cell phenotypes in numerous solid tumors[2], including breast[3], ovarian[4], pancreatic[5] and brain cancers[6]. However, sparse and aligned collagen fibers at the edges of tumors have also been reported to correlate with aggressive disease[7]. It remains unclear whether and how collagen architectures have a role in driving metastatic migration programs or if they simply correlate with progression of the tumor.

Intravital microscopy studies have shown that distinct collagen architectures are associated with specific cell motility behaviors. Cancer cells migrating through densely packed collagen within the tumor use invadopodia and matrix metalloproteinase (MMP) activity to move, whereas cells in regions with less dense collagen and long, aligned fibers migrate rapidly using larger pseudopodial protrusions or MMP-independent ameboid blebbing[8, 9]. Likewise, we previously showed in vitro that cell migration speed, invasion distance, and cellular protrusion dynamics are modulated by collagen fiber alignment, but that this relationship breaks down at high collagen densities (>2.5 mg ml$^{-1}$)[10]. These findings suggest that distinct motility regimes exist in low-density and high-density collagen, which may have implications for metastatic progression.

Here, we explore the relationships between collagen density, collagen architecture, cell migration behavior, gene expression, and metastatic potential. To do this, we develop a 3D in vitro model system designed to probe the physical basis of cancer cell migration responses to collagen matrix organization. Using this system, we find that confining collagen matrix architectures with short fibers and small pores induce a conserved migration behavior in cancer cells leading to network formation and the upregulation of a conserved transcriptional module, both of which are mediated by integrin-β1 upregulation. We show evidence that this in vitro behavior is consistent with phenotypic and molecular features of clinical VM. Moreover, we show that the associated transcriptional response is conserved among cancer types in vitro and is predictive of patient survival in multiple clinical datasets for various tumor types. Our integrative study suggests that a collagen-induced migration phenotype and gene

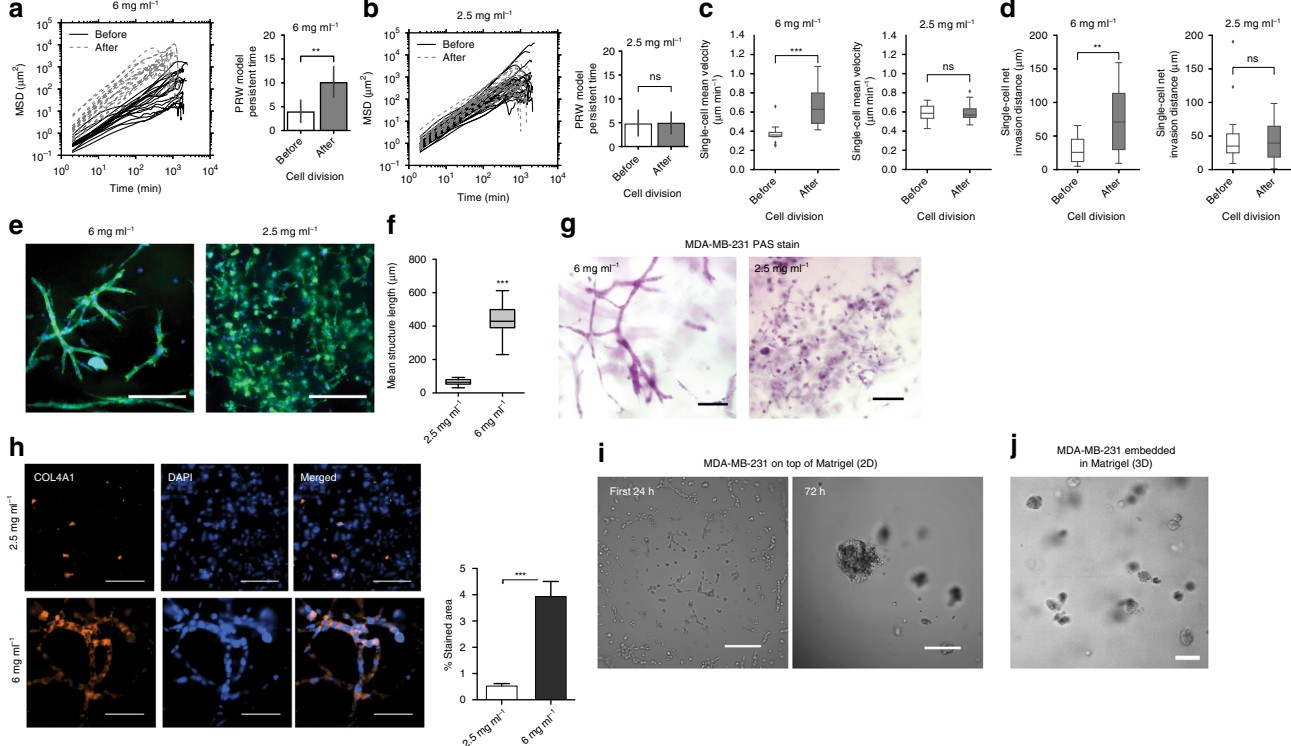

**Fig. 1** High-density 3D collagen microenvironment promotes a switch to persistent cell migration in cancer cells. **a** Mean squared displacement (MSD) and persistent time of MDA-MB-231 cells before and after cell division in high-density collagen. The persistent time was calculated from the MSDs using the persistent random walk model (see "Methods"). MSDs are shown for 12 representative cell trajectories. **b** Mean MSD and persistent time of MDA-MB-231 cells before and after cell division in low-density collagen. The persistent time was calculated from the MSDs using the persistent random walk model (see "Methods"). MSDs are shown for 12 representative cell trajectories. **c** Single-cell velocity measured at 2 min intervals before and after cell division. **d** Single-cell net invasion distance before and after cell division for cells in high-density and low-density collagen. **e** Representative image of MDA-MB-231 cells cultured in a 6 mg ml$^{-1}$ (left) and in a 2.5 mg ml$^{-1}$ (right) collagen I matrix after 7 days of culture. Cells are stained with Alexa-488 Phalloidin (F-Actin) and DAPI (nuclei). Scale bar 250 μm. **f** Quantification of mean structure length in low-density and high-density collagen, from images acquired in three independent experiments. **g** PAS stain of MDA-MB-231 cells cultured for 7 days in a 3D collagen gel of high-density (left) and low-density (right). Scale bar 100 μm. **h** Immunofluorescence staining of MDA-MB-231 cells for collagen IV after 7 days of culture in 6 vs. 2.5 mg ml$^{-1}$. Representative images of $n = 2$ biological replicates. Bar graph shows mean and s.e.m. of quantification of stained area performed in 15 different fields of view. Scale bar 100 μm. **i** MDA-MB-231 cells cultured on top of growth factor-reduced matrigel after 24 h (left) and after 72 hours (right). Scale bar 250 μm. **j** MDA-MB-231 cells cultured inside growth factor-reduced matrigel in 3D culture for 7 days. Scale bar 100 μm. Box plots show quartiles of the dataset with whiskers extending to first and third quartiles. $n = 3$ biological replicates for all experiments unless otherwise noted. Statistical significance was determined by Mann–Whitney $U$ test and is indicated as *, **, *** for $p \leq 0.05$, $p \leq 0.01$, $p \leq 0.001$, respectively

expression program are linked to a metastatic clinical tumor cell phenotype and potentiates future work to identify mechanistic strategies capable of limiting metastasis in several cancers.

## Results

**High-density collagen promotes fast and persistent migration**. To first investigate the role of 3D collagen density in modulating the migration phenotype of breast cancer cells, we embedded MDA-MB-231 cells in collagen I matrices at densities mimicking normal breast tissue, 2.5 mg ml$^{-1}$ collagen[10, 11], and cancerous breast tissue, 6 mg ml$^{-1}$ collagen[10, 11]. We observed that cells migrating in dense collagen initially appeared to be trapped and were unable to invade. However, after one division cycle, most cells switched to a highly invasive motility behavior, significantly increasing their persistence, velocity, and total invasion distance (Fig. 1a–d, left panels). This behavior was not observed in cells embedded in the low-density matrix, where cell migration was the same before and after division (Fig. 1a–d, right panels). Interestingly, cells that were in contact with the coverslip and not fully embedded in the high-density condition did not undergo the same migration transition upon division (Supplementary Fig. 1a, b). The motility responses we observed in 2.5 and 6 mg ml$^{-1}$ collagen matrices were not unique to MDA-MB-231 breast cancer cells. Similar migration patterns were observed for HT-1080 fibrosarcoma cells embedded in the same collagen matrix conditions (Supplementary Fig. 1c), suggesting that these responses may be conserved among distinct cancer types. To further examine whether the observed migration behavior was cell-type dependent, we tested the response of normal mesenchymal human foreskin fibroblasts (HFF-1) to low-density and high-density collagen conditions. Over an observation period of 48 h, HFF cells migrated consistently with very low persistence. Cells invaded less than three cell lengths in low-density collagen. In high density, HFFs elongated to reach cell lengths up to 300 μm but did not invade significantly (Supplementary Fig. 1d).

**Density-induced migration results in cell network structures**. It was unexpected that both MDA-MB-231 and HT-1080 cancer cells migrated faster and further in high-density collagen conditions. Intuitively, cell migration would be expected to slow in dense conditions where more matrix must be remodeled to enable cell movement. Moreover, this behavior was common to both cancer cell types but not displayed by normal fibroblasts, which represent residents of the stroma and also undergo mesenchymal migration in collagen. This motivated us to investigate the long-term implications of the rapid migration phenotype induced in cancer cells under high-density conditions. After 1 week of culture in high-density collagen, breast cancer cells undergoing rapid, and persistent migration formed interconnected network structures that resembled the early stages of endothelial tubulogenesis (Fig. 1e, left). The average length of cell networks after 1 week was 437 μm (Fig. 1f). Interestingly, these network structures do not appear to be caused by cells aligning along collagen fibers (Supplementary Fig. 1e). In contrast, cells cultured in low-density collagen for 1 week migrated slowly with low persistence, and remained as single cells (Fig. 1e, right). HT-1080 cells also formed network structures in high-density collagen and remained as single cells in low-density collagen (Supplementary Fig. 1f). HFFs remained as single cells in both high-density and low-density conditions (Supplementary Fig. 1g). The transition of cancer cells from single-cell migration to network formation suggested a potential transdifferentiation event, and the cell networks were reminiscent of a cancer phenotype known as vasculogenic mimicry (VM). VM is thought to arise from tumor cells that

acquire the ability to form networks in the tumor ECM lined with glycogen-rich molecules and basement membrane proteins that can be perfused with blood. However, the tumor cells lining these networks do not express endothelial surface markers such as CD31[12, 13]. Periodic acid schiff (PAS) staining of the networks formed in our high-density collagen condition confirmed the presence of glycogen-rich molecules (Fig. 1g) and immuno-fluorescence confirmed the presence of basement membrane protein COL4A1 (Fig. 1h), as in VM.

Previous pioneering studies have shown that several aggressive melanoma cell lines, which produce VM in vivo also intrinsically form VM network structures when cultured on top of Matrigel or collagen I in a 2D in vitro context[13, 14]. Recently, other aggressive tumor cell types have been shown to intrinsically form VM-like network structures on top of Matrigel or in 2.5D culture in Matrigel[15–19]. Here, it is important to note that variations exist in the consistency of commercial ECM products as well as the terminology used to describe 3D culture. We define 3D culture strictly as a condition where cells are fully embedded, in contact with ECM on all sides, and located a sufficient distance away from the coverslip bottom and sides of the culture dish to avoid their influence. We define 2.5D culture as a pseudo 3D culture where cells are embedded in the ECM but in contact with coverslip. Our previous studies have demonstrated the importance of these distinctions, as cell behavior and protein localization are differentially regulated in each context[20–22]. Therefore, we sought to understand whether the network phenotype induced by a 3D collagen I environment was distinct from that induced by a 2D Matrigel environment. First, we asked whether our cells formed network structures on top of Matrigel. Few cells aligned within the first 24hrs of culture, and nearly all cells aggregated after 72 h (Fig. 1i). Next, we embedded MDA-MB-231 cells inside of Matrigel, in 3D culture. In this context, cells did not form network structures but instead formed rough-edged, disorganized spheroids (Fig. 1j). Thus, high-density collagen uniquely induced the network-forming phenotype in a more physiologically relevant 3D context.

**A conserved transcriptional response precedes migration**. We hypothesized that the persistent migration phenotype of cancer cells leading to network formation in high-density collagen conditions (collagen-induced network phenotype, CINP) could be the result of a transdifferentiation event wherein a unique cell motility gene module was upregulated. To test this, we conducted RNA sequencing of MDA-MB-231, HT-1080, and HFF cells cultured in low-density and high-density collagen matrices after 24 h (Fig. 2a), the time point just before most cancer cells in the high-density collagen matrix underwent at least one cycle of cell division and began to invade with increased persistence. As the majority of cancer cells cultured under high-density conditions participated in network formation, we expected their bulk transcriptional profile to be dominated by this phenotype[23]. Then we asked whether common stem cell and differentiation markers were upregulated in association with the network-forming phenotype. Indeed, several known stem cell markers were upregulated (Fig. 2b), and three were common to both cancer cell types: JAG1, ITGB1, and FGFR1. This suggested that both cancer cell types harbored stem-like qualities, which could facilitate significant transcriptional reprogramming.

Analyzing more broadly, we then asked which genes were differentially regulated (TPM fold change ≥1.5) in high-density collagen compared to low-density collagen in each cell type and whether these genes represented unique or conserved transcriptional response modules. As expected, cell type accounted for the most variance in gene expression (Fig. 2c). However, after a

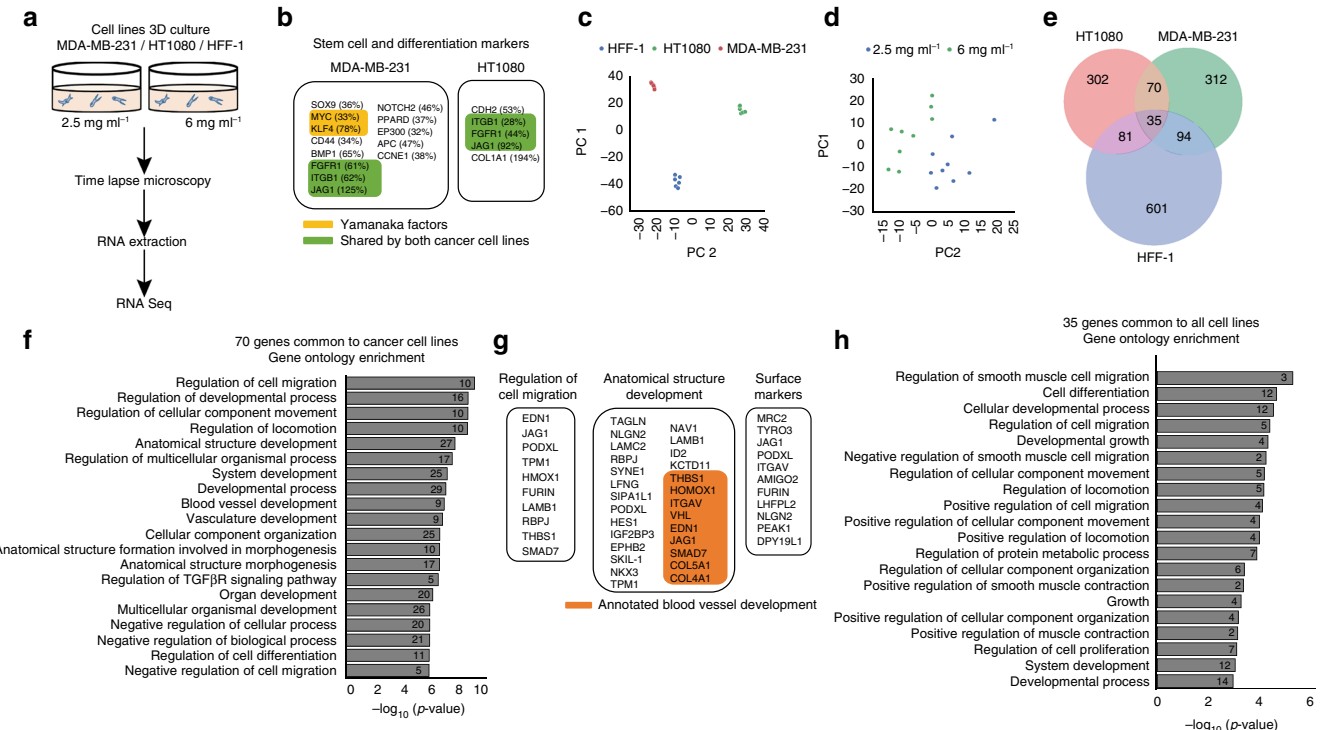

**Fig. 2** The network-forming phenotype induced by high-density 3D collagen is accompanied by a transcriptional response common-to-cancer cells. **a** Schematic of the experimental approach. Each cell line in each condition was cultured in biological triplicate, and each replicate was sequenced ($n = 3$ for each cell type per condition). **b** List of genes upregulated in each of the cancer cell lines that are known stem cell or differentiation markers. **c** Principal component analysis of raw RNA-Seq data shows cell type as main driver of variance in gene expression. **d** Principal component analysis of z-score transformed data shows culture condition as the main driver of variance in gene expression. **e** Venn diagram showing the overlap between genes upregulated in 6 vs. 2.5 mg ml$^{-1}$ collagen in the three cell lines analyzed. **f** Gene ontology (GO) of biological processes enriched in the 70 genes upregulated by cancer cells in 6 mg ml$^{-1}$ collagen. Number at the end of the bars represents number of genes annotated for the particular GO term. **g** Lists of genes with annotations relevant to the observed phenotype. Left: regulation of cell migration. Middle: regulation of anatomical structure development. Orange color highlights genes annotated for blood vessel development. Right: surface markers. **h** GO of biological processes enriched in the 35 genes shared by cancer cells and HFF-1 fibroblasts. Number at the end of the bars represents number of genes annotated for the particular GO term. See "Methods" for analysis details

z-score transformation of the gene expression of each cell type, the collagen matrix condition accounted for the bulk of the remaining variance in gene expression (Fig. 2d). This suggested the presence of gene expression programs linked to collagen matrix conditions.

Using a Venn diagram approach to identify conserved expression modules, we discovered a set of 70 genes that were upregulated by both cancer cell types but not normal cells in response to high-density collagen (Fig. 2e; Supplementary Fig. 2a). Gene ontology (GO) enrichment analysis revealed that these 70 common-to-cancer genes were significantly enriched for annotations in blood vessel development and regulation of migration (Fig. 2f, g). Importantly, changes in the threshold for differential expression did not significantly alter the primary gene ontology categories identified (Supplementary Fig. 2d; Supplementary Table 1). Key genes involved in Notch signaling, i.e., *RBPJ* and *LFNG*, were among the 70. Importantly, *LAMC2*, *JAG1*, and *THBS1* genes identified in this common-to-cancer gene set have been previously associated with a VM phenotype intrinsically displayed by metastatic melanoma, which was assessed by targeted microarray analysis for angiogenesis, ECM, and cell adhesion genes[24, 25]. Upregulated surface markers were not endothelial in nature, and did not represent any specific tissue or cell type (Fig. 2g).

Further exploration of our dataset with respect to individual cancer cell types revealed that, beyond the conserved transcriptional response, high-density collagen also triggered the expression of genes related to vasculogenesis in a cell type-dependent manner. For example, breast cancer cell networks upregulated *VEGFA* fold change = 1.65 and *MMP14* fold change = 1.72, but fibrosarcoma cell networks did not. Some of these genes have been previously associated with the VM network phenotype of melanoma cells (Supplementary Fig. 2c)[24].

Next, we assessed the 35 genes that were upregulated in response to high-density collagen by all three cell types (Fig. 2e). These genes were enriched primarily for annotations in regulation of cell differentiation (Fig. 2h). However, it is important to take into account the inherent flaws associated with GO enrichment analysis. For example, some categories showing enrichment in the 35 genes common-to-all cell lines contain very few genes and may not represent real enrichment. However, this limitation is not observed in the top enriched categories in the 70 genes common-to-cancer cells, where most category contains at least 10 genes (Fig. 2f). The genes associated with each enrichment category are given in Supplementary Tables 2 and 3.

Interestingly, *SERPINE1*, a secreted protease inhibitor involved in coagulation and inflammation regulation, was identified in the common-to-all gene module (Supplementary Fig. 2b). Several Serpine protein family members have previously been implicated

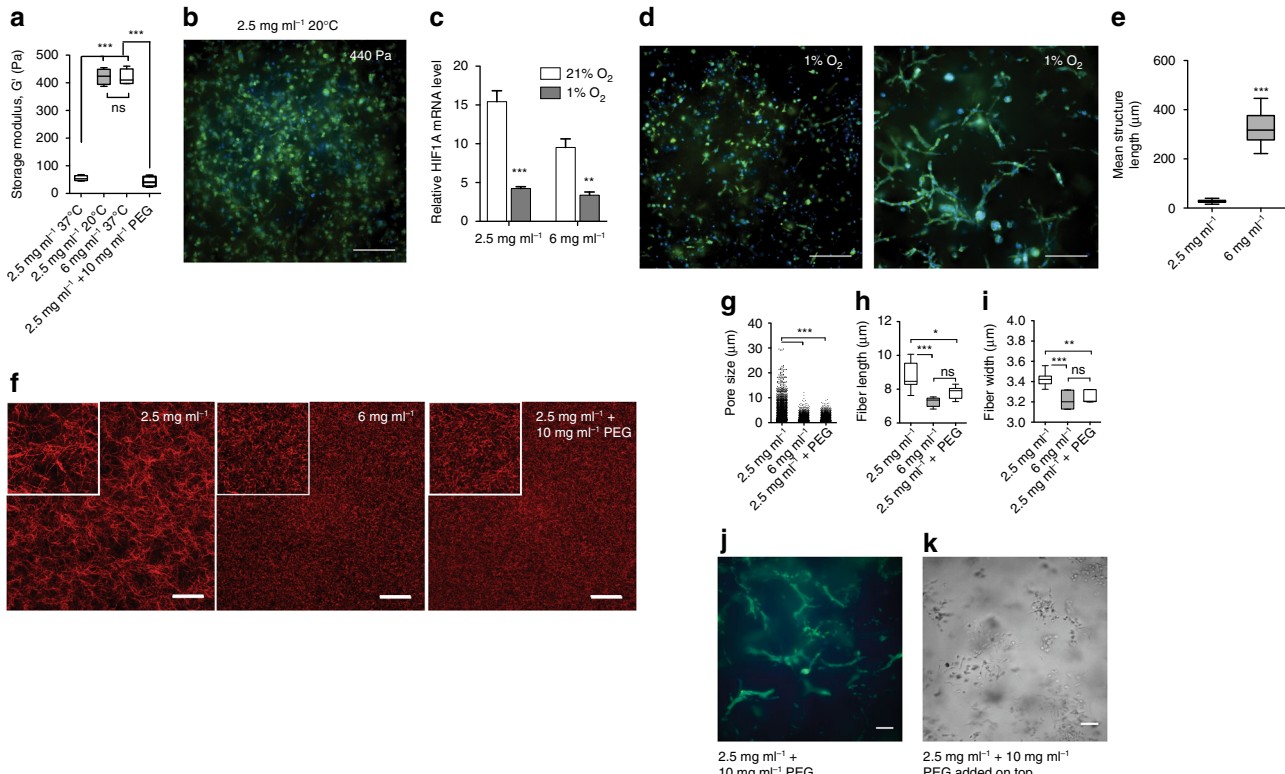

**Fig. 3** Cell network formation is not triggered by hypoxia or matrix stiffness but rather by matrix architecture. **a** Storage modulus of collagen gels as estimated by shear rheology after polymerization at different temperatures. **b** Representative images of cells after 7 days of culture in low-density collagen polymerized at 20 °C (high stiffness, 440 Pa). **c** *HIF1A* expression in low-density and high-density 3D collagen after 7 days of culture under normoxic (21% $O_2$) or hypoxic (1% $O_2$) conditions. **d** Representative images of MDA-MB-231 cells in low-density and high-density 3D collagen after 7 days of culture under hypoxic (1% $O_2$) conditions, scale bar 250 μm. **e** Quantification of mean structure length after 7 days of culture under hypoxic (1% $O_2$) conditions in low-density and high-density collagen. **f** Confocal reflection images of collagen fibers in 3D matrices. Left: 2.5 mg ml$^{-1}$ collagen I, center: 6 mg ml$^{-1}$ collagen I, and right: 2.5 mg ml$^{-1}$ collagen + 10 mg ml$^{-1}$ PEG. Insert shows a ×2 zoom. Scale bar 100 μm. **g** Quantification of pore size in the three conditions showed in **f**. **h** Fiber length and **i** fiber width as measured from the confocal reflection images in the three conditions showed in **f**. **j** Representative image of MDA-MB-231 cells cultured for 7 days in a 2.5 mg ml$^{-1}$ collagen + 10 mg ml$^{-1}$ PEG 3D matrix. Cells are stained with Alexa-488 Phalloidin (F-Actin) and DAPI (nuclei). Scale bar 250 μm. **k** Representative bright field image of MDA-MB-231 breast cancer cells cultured in a 2.5 mg ml$^{-1}$ collagen matrix where 10 mg ml$^{-1}$ PEG was added to the media after polymerization. Scale bar 125 μm. Bar graphs represent mean ± s.d. and data in box and whiskers plots is presented using Tukey method. $n = 3$ biological replicates for all experiments unless otherwise noted. Statistical significance was determined by ANOVA (**a**, **c**, **g**–**i**) and Mann–Whitney U test (**e**) and is indicated as *, **, *** for $p \leq 0.05$, $p \leq 0.01$, $p \leq 0.001$, respectively. Bars plots are mean ± s.d

as drivers of metastasis correlating with VM[18] and with brain metastases of lung and breast cancers[26].

**Integrin-β1 upregulation is required for CINP**. We next sought to identify the matrix feature triggering transdifferentiation. The physical parameters of stiffness, pore size, and fiber organization differ between the low density 2.5 mg ml$^{-1}$ and high density 6 mg ml$^{-1}$ collagen matrices[10]. Chemical cues may also change. For example, adhesive ligand density and binding site presentation to integrins and other matrix receptors may differ[27, 28] as well as accumulation or release of autocrine and paracrine signals sequestered by the ECM[29–31]. Each of these features could potentially impact cancer cell motility behavior and gene expression.

As matrix stiffness has been implicated in driving epithelial-to-mesenchymal transitions (EMT) and aggressive phenotypes[11, 32, 33], we first asked whether increased stiffness of the high-density collagen matrix was responsible for triggering transdifferentiation. To test this, we developed a collagen polymerization procedure ("Methods") that increases the stiffness of the low-density matrix to match the stiffness of the high-density matrix (Fig. 3a). By lowering the polymerization temperature from 37 to

20 °C, polymerization slowed, allowing fibers to form more organized and reinforced fiber structures with larger pores (Supplementary Fig. 1I). Breast cancer cells cultured in this stiffened low-density condition did not undergo network formation (Fig. 3b), suggesting that 3D stiffness is not sufficient for triggering the transdifferentiation.

Next, we sought to determine whether the smaller pore size of the high-density matrices triggered transdifferentiation. One way in which smaller pore sizes could influence cell behavior is by restricting the diffusion of molecules to and from the cells[34]. More specifically, the imbalance between oxygen diffusion to cells and oxygen consumption by cells in 3D matrices has been shown to promote hypoxic conditions in some cases[35]. Since regions of VM have previously been associated with markers of hypoxia in vivo[36, 37], we hypothesized that cells in high-density collagen created a more hypoxic condition than in low-density collagen and that low-oxygen levels could trigger network formation. To test this, we cultured MDA-MB-231 cells in low-density collagen under a hypoxic atmosphere of 1% oxygen for 1 week. To confirm that a hypoxic response was achieved, we assessed the level of *HIF1A* mRNA expression by RT-qPCR at day 7 and found a significant decrease in *HIF1A* expression (Fig. 3c). This is a common response to long-term hypoxia by various cancer cell

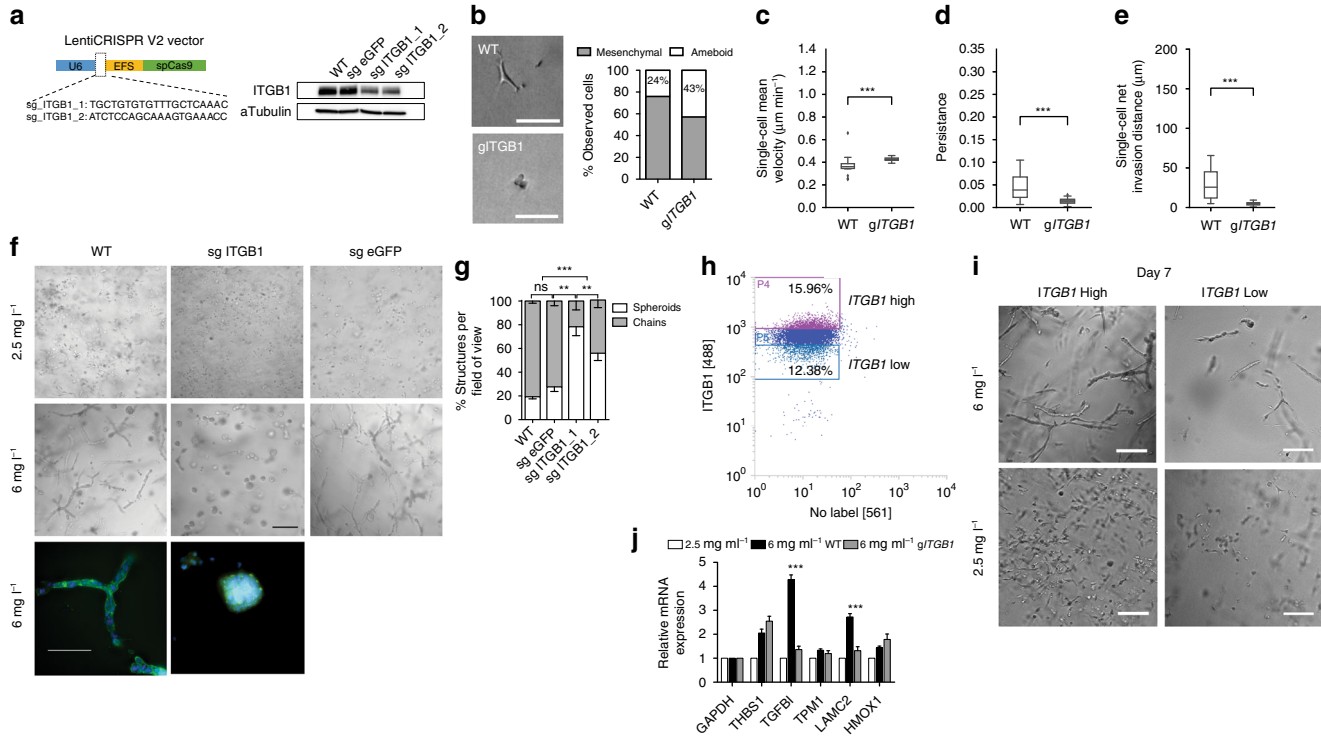

**Fig. 4** Role of β1 integrin in the formation of cell network structures in high-density collagen. **a** Schematic of lentiCRISPR V2 vector used for targeting *ITGB1* gene and western blot validation of the protein depletion after 7 days of cell transduction. **b** Comparison of MDA-MB-231 cells WT and *ITGB1* depleted in low-density 3D collagen. Left: micrographs showing a representative image of a WT cell undergoing mesenchymal migration and an *ITGB1*-depleted cell undergoing ameboid migration. Right: quantification of mesenchymal vs. ameboid migration within the cell populations. **c** Quantification of the effect of *ITGB1* depletion on mean cell velocity when cells are cultured in 6 mg ml$^{-1}$ collagen. **d** Cell persistence and **e** cell invasion distance. Comparison for **c**–**e** was performed using Mann–Whitney *U* test. **f** MDA-MB-231 WT, *ITGB1*-depleted, and control sgRNA cell phenotypes after 7 days of culture in low-density collagen (top row) and high-density collagen (middle row). Scale bar 250 μm. Bottom row shows high-magnification micrographs highlighting the difference between chain structures and spheroids. Scale bar 100 μm. **g** Quantification of proportional number of structures in each cell line when cultured in high-density collagen. **h** Fluorescence-activated cells sorting (FACS) was used to separate the parental WT MD-MB-231 cell line population into high-*ITGB1* and low-*ITGB1* expressing populations. **i** *ITGB1* high and *ITGB1* low cells after 7 days of culture in high-density 3D collagen (top row) and low-density (bottom row). Scale bar 200 μm. **j** RT-qPCR quantification of a small subset of genes identified in the 70-gene module in WT control- and *ITGB1*-silenced cells when cultured in low-density and high-density collagen. Data show mRNA levels relative to GAPDH and relative to low-density collagen level. Statistical significance evaluated between WT and g*ITGB1* groups, Statistical significance was determined by ANOVA test. Bar graphs represent mean ± s. d. and data in box and whiskers plots is presented using Tukey method. *n* = 3 biological replicates for all experiments unless otherwise noted. Significance is indicated as *, **, *** for *p* ≤ 0.05, *p* ≤ 0.01, *p* ≤ 0.001, respectively

lines[38–40]. However, hypoxia was not sufficient to induce network formation in any portion of the cancer cell population in the low-density collagen matrix (Fig. 3d, left). For comparison, we also assessed the *HIF1A* mRNA expression of breast cancer cells cultured for 1 week in low-density collagen under 21% oxygen, in high-density collagen under 1% oxygen, and in high-density collagen under 21% oxygen (Fig. 3c). These results suggested that cells cultured in high-density collagen experience increased hypoxia compared to cells cultured in low-density collagen under normal atmospheric conditions. Nevertheless, the hypoxic response achieved in low-density collagen under 1% oxygen exceeded that induced by high-density matrix alone. Cells in high-density matrix under 1% oxygen continued to predominately display a network phenotype (Fig. 3d, right), but the average network length (Fig. 3e) was significantly shorter than cells in high-density collagen under normoxic conditions (Supplementary Fig. 1h). Previous studies have reported that hypoxia is not sufficient to induce a VM phenotype in melanoma cells in vitro[13]. It is possible that in vivo, additional stromal cell secreted factors or cell–cell interactions modulated by hypoxia may indirectly influence the VM process[37, 41].

To further explore whether pore size reduction induced transdifferentiation of cancer cells, we sought to interrogate this parameter independently of collagen density. In our model, the high-density condition contains 2.4 times more collagen than the low-density condition. This increase in total collagen reduces pore size, but also presents more adhesive ligands to cells, which could increase integrin activation. To separate pore size from bulk density, we developed a collagen structure engineering technique that reduced the pore size and fiber length of the low-density matrix to approximate that of the high-density matrix. Under normal polymerization conditions, low-density collagen self-assembles into relatively long, structured fibers. When non-functionalized, inert polyethylene glycol (PEG) was mixed into collagen monomer solution prior to polymerization, molecular crowding-restricted fiber formation. This resulted in shorter, more interconnected fibers yielding smaller pores (Fig. 3f–i) without increasing stiffness (Fig. 3a). Breast cancer cells encapsulated in this pore size-reduced low-density matrix underwent network formation over the course of 1 week (Fig. 3j). To control for the possible influence of PEG itself, PEG was added into media on top of a normally polymerized low-density gel embedded with cells and allowed to diffuse into the interstitial

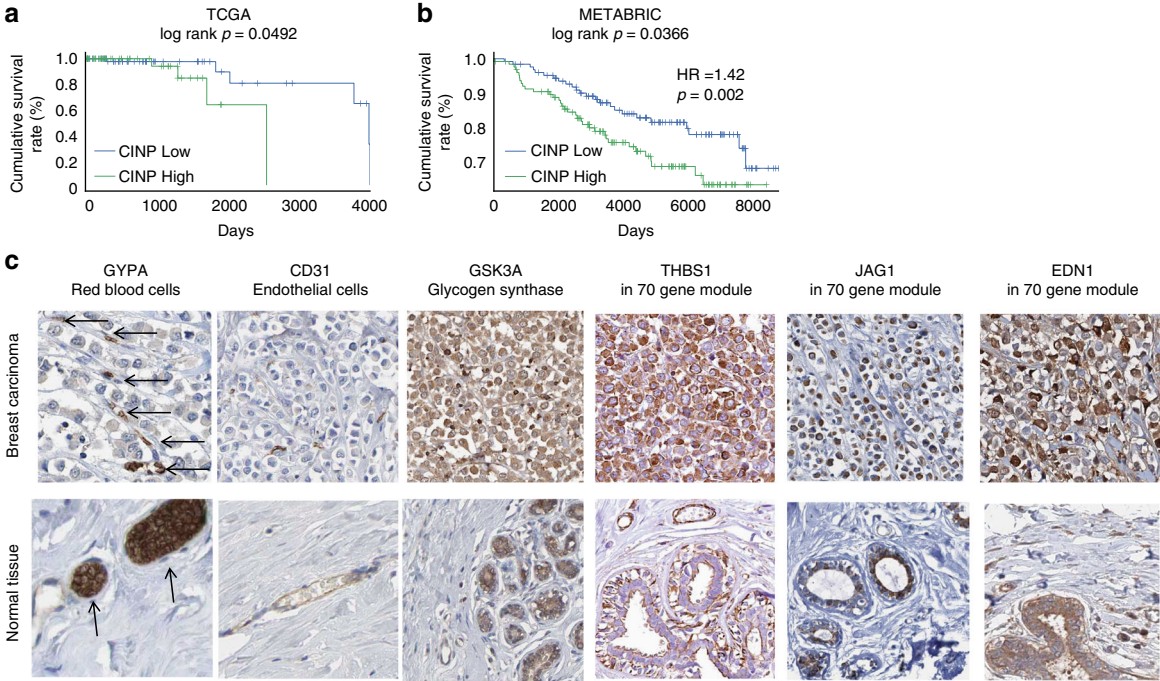

**Fig. 5** The transcriptional response module associated with the collagen-induced network phenotype (CINP) is predictive of poor prognosis in human tumor datasets. **a** Kaplan–Meier survival analysis of stage I breast cancer patients from TCGA and **b** METABRIC databases, when the PC1 loadings were used as an expression metagene. High CINP refers to the highest metagene expression scores and low CINP to the lowest expression scores. HR indicates hazard ratio. **c** Sections of a primary breast carcinoma displaying the clinical VM phenotype of chain-like cell structures surrounded by a matrix network. Column 1: red blood cells, stained by an antibody against GYPA, are indicated by arrows. Several red blood cells are traversing the matrix surrounded by cancer cells. Column 2: tumor cells are negative for CD31 but in healthy tissue, stained regions colocalize to vessel structures. Column 3: tumor cells stain strongly for glycogen synthase, which likely contributes to the generation of a glycogen-rich matrix between the chains of cells. Columns 4–6: tumor cells undergoing VM stain strongly for three of the most upregulated genes in our 70-gene module. Image credit for **d**: Human Protein Atlas, patient ID 1910, available from www.proteinatlas.org[48]. See "Methods" for analysis details

spaces among the fibers to reach the same final concentration as was used in the pore size-reduced low-density matrix (10 mg ml$^{-1}$ PEG). Cells maintained in this molecularly crowded condition over 1 week did not form networks, but instead remained as single cells. However, a noticeable slowing of cell migration occurred, which resulted in an anisotropic patterning of single cells throughout the matrix (Fig. 3k). These results suggested that the fiber architecture of high-density collagen induces network formation independently of the bulk increase in adhesive ligand density and confirms that bulk matrix stiffness is not involved.

The short, more isotropic arrangement of fibers associated with both the high-density collagen and low-density PEG crowded collagen conditions could act on cells through local cell–matrix interactions transduced by integrin signaling. Integrin-β1 (*ITGB1*) is a canonical receptor for collagen I, a central node in ECM signal transduction, and a critical mediator of breast cancer progression in mouse and in vitro models[42]. Here, *ITGB1* was upregulated by both cancer cell types in response to confining matrix conditions (Fig. 2b). Thus, we next asked whether the network-forming phenotype observed in confining matrix conditions was mediated by *ITGB1*. CRISPR-Cas9 technology was used to silence *ITGB1* expression with single guide RNAs (sgRNAs), and constructs expressing sgRNAs targeting eGFP were used as controls (Fig. 4a). Silenced and control cells were embedded separately and sparsely in low-density and high-density collagen matrices. Cells were monitored by time-lapse microscopy for early migration behavior then imaged again after one week. In low-density collagen, *ITGB1*-silenced cells maintained a similar level of migration capability to wild type (WT) cells in low-density matrices, but used an ameboid blebbing

migration phenotype instead of a mesenchymal migration phenotype (Fig. 4b). In high-density conditions, *ITGB1*-silenced cells migrated faster than WT cells, but were significantly less persistent and did not invade (Fig. 4c–e). Surprisingly, after 1 week *ITGB1*-silenced cells in high-density collagen-formed spheroid structures instead of cell networks, whereas control cells exhibited the same behavior as the WT in both collagen conditions (Fig. 4f). Retrospective analysis of WT MDA-MB-231 cells in high-density collagen revealed that a small fraction spontaneously formed spheroid structures (Fig. 4g). These findings suggest that either basal expression level or upregulation of *ITGB1* dictates the network-forming phenotype. To distinguish between these two possibilities, we next sorted the parental WT population based on basal *ITGB1* expression level and then embedded high and low expressing cells separately in confining high-density collagen matrices (Fig. 4h). We observed no appreciable differences in the percentage of networks versus spheroids formed by the sorted populations after one week. However, *ITGB1* low cells proliferated less and displayed fewer total number of network or spheroid structures (Fig. 4i) even though the initial seeding density was the same (Supplementary Fig. 3a).

To further explore the link between the upregulated transcriptional module and the network-forming phenotype, we asked whether *ITGB1*-silenced spheroid-forming cells showed different gene expression patterns than WT network-forming cells. To assess this, we conducted qRT-PCR analysis of a subset of the 70-gene panel in the two cell phenotypes. Upregulation of several key genes were maintained in the spheroid-forming cells, whereas other genes were no longer upregulated (Fig. 4j). These results

**Table 1 CINP score potential to predict prognosis in stage I patients from metabric database broken down by molecular subtype**

| Metabric molecular subtype | Patient count | Death observed | HR | Cox p |
|---|---|---|---|---|
| Luminal B | 126 | 33 | 1.2461 | 0.3194 |
| Luminal A | 202 | 34 | 1.5996 | 0.0162 |
| Triple negative | 63 | 14 | 3.8537 | 0.0070 |
| HER2+ | 39 | 13 | 0.7152 | 0.3405 |

Analysis of CINP score potential to predict prognosis in stage I patients from METABRIC database broken down by molecular subtype

**Table 2 TCGA pan cancer analysis independent of stage**

| Cancer type | Patient count | Death observed | HR | Cox p |
|---|---|---|---|---|
| LGG | 508 | 92 | 1.8434 | 1.1E−13 |
| ACC | 79 | 25 | 3.1863 | 2.8E−04 |
| CESC | 304 | 60 | 1.6560 | 5.2E−04 |
| MESO | 85 | 28 | 1.6101 | 6.9E−04 |
| PAAD | 178 | 59 | 1.5948 | 2.2E−03 |
| BLCA | 409 | 111 | 1.3338 | 0.0053 |
| LUAD | 521 | 124 | 1.2448 | 0.0169 |
| KICH | 64 | 8 | 2.9277 | 0.0210 |

Table showing results from Kaplan–Meier and hazard ratio analysis across all cancer types in TCGA, where the CINP gene score is significant predictor of prognosis ($p < 0.05$)

show that *ITGB1* regulates some aspects of the transcriptional module associated with the network-forming phenotype.

Finally, we asked whether upregulated genes in our transcriptional module that have previously been implicated as drivers of VM in vitro were functionally active in our network-forming phenotype. *LAMC2* (Ln-5, gamma 2 chain) was previously found to be upregulated in aggressive melanoma cells that intrinsically display the VM phenotype compared to less aggressive melanoma cells that do not display VM. Moreover, it was implicated as a driver of VM network formation, since the cleavage of this secreted matrix molecule by *MMP-2* and *MT1-MMP* produces pro-migratory fragments. In 2D culture of aggressive melanoma cells on top of collagen I, the inhibition of *LAMC2* cleavage blocked VM network formation[43]. Using shRNA to knockdown *LAMC2*, we found that *LAMC2* KD MDA-MB-231 cells maintain their ability to form network structures in 3D high-density collagen (Supplementary Fig. 3b, c). *COL4A1* is another matrix molecule upregulated by cells undergoing the network phenotype (Figs. 1h and 2g) and previously implicated in driving migration[44]. *COL4A1* KD in MDA-MB-231 cells also did not inhibit the ability of cells to form network structures in 3D high-density collagen (Supplementary Fig. 3b, c).

**CINP transcriptional module predicts poor prognosis in human cancer**. Finally, we sought to determine whether the CINP triggered by our 3D system was clinically relevant. To test this, we first asked whether the 70 common-to-cancer genes associated with the CINP could predict cancer patient prognosis. We anticipated that if this gene signature was indicative of a more metastatic cancer cell migration phenotype, its expression would correlate with poor patient outcomes. Since late stage tumors are already characterized by migration of tumor cells to distant lymph nodes or organs, we hypothesized that a gene signature associated with metastatic migration would correlate with prognosis in early (stage I and II) but not late (stage III and IV) stage tumors. Using the cancer genome atlas (TCGA), we first analyzed data for breast cancer patients with respect to the expression of the 70-gene signature. An expression metagene was constructed using the loadings of the first principal component (CINP PC1) of a 195 Stage I patient by 70-gene matrix (Supplementary Fig. 4a, also see "Methods"). Then a survival analysis was conducted, comparing patients with the highest (top 30%) and lowest (bottom 30%) expression metagene scores by log-rank test. The cumulative survival rate of these two groups differed significantly (log-rank $p = 0.049$); however, there was insufficient data to power a hazard ratio (HR) calculation (Fig. 5a). Analysis using the more data-rich METABRIC microarray database of breast cancer patients showed similar results for Stage I, confirming the prognostic value of the gene set (log-rank $p = 0.037$, HR = 1.40, Cox $p = 0.002$, Fig. 5b). Applying the same analysis to stage II

breast cancer patients revealed that the CINP metagene was associated with a marginally significant difference in 5-year survival by TCGA analysis but not by METABRIC analysis (Supplementary Fig. 4b, c). One caveat to this analysis is that data for 11 of the genes in our 70-gene panel were not available in the METABRIC dataset. The CINP metagene also did not separate patients with better prognosis in late stage tumors (Supplementary Fig. 4d). These results indicate that the CINP gene module could have clinical predictive power in the early stages of breast cancer. Importantly, further analysis of stage I patients by molecular subtype[45] revealed that the CINP metagene provided significant prognostic value for Luminal A and triple negative breast cancer patients (Table 1).

Next, we screened the predictive value of the gene module in additional cancer types in TCGA independently of stage or subtype using only age and CINP score as covariates. The CINP gene module was a significant predictor of survival in lung adenocarcinoma (LUAD), lower grade glioma (LGG), cervical squamous cell carcinoma and endocervical adenocarcinoma (CESC), pancreatic adenocarcinoma (PAAD), mesothelioma (MESO), adrenocortical carcinoma (ACC), bladder urothelial carcinoma (BLCA), and kidney chromophobe carcinoma (KICH) (Table 2), but was not a significant predictor in several other tumor types found in TCGA (Supplementary Table 3).

Finally, we sought to determine whether the in vitro network-forming phenotype and associated transcriptional signature were related to the clinical VM phenotype. Using the Human Protein Atlas (www.proteinatlas.org)[46], we first identified breast cancer tumor slices displaying hallmarks of the VM phenotype, namely linear chains of cells lining glycogen-rich matrix networks that conduct blood flow but do not stain positively for *CD31*[13]. The tumor of patient 1910 displayed linear chains of cancer cells lining interconnected matrix networks (Fig. 5c). An immunohistochemical stain for *GYPA* showed red blood cells flowing through the matrix networks in tumor tissue but highly concentrated in vessel-like structures in healthy tissue. A stain against *CD31* showed that there were no endothelial cells lining the matrix networks in the tumor tissues. Although a PAS stain was not available in the protein atlas database, which would determine whether the matrix networks were positive for glycogen, a stain against glycogen synthase (*GSK3A*) was available and showed that the chains of cancer cells significantly expressed this enzyme. The network-forming cell phenotypes combined with IHC evidence are consistent with the previously described histopathology of VM[13]. Next, we asked whether highly upregulated genes in our 70-gene CINP module were evident at the protein level in this clinical sample of VM. Stains for *THBS1*, *JAG1*, and *EDN1* were available in the protein atlas database for the same tumor and showed significant expression of all three

genes from our CINP transcriptional module in the VM tumor tissue but little stain in healthy tissues.

## Discussion

Our transcriptional, histopathologic, and phenotypic data suggest that the in vitro CINP and clinical VM share many commonalities. To our knowledge, this is the first time that collagen fiber architecture, characterized by short fibers and small pores, has been identified as an inducer of cancer transdifferentiation associated with a VM-like phenotype or more normal acinar phenotype, depending on the capacity of cells to upregulate *ITGB1*. More broadly, our findings show that collagen fiber architecture modulates the role *ITGB1* plays in migration. In one architectural context, *ITGB1* facilitates a switch from mesenchymal to ameboid migration and in another architectural context it mediates migration persistence and the shape of structures formed by collective morphogenesis.

Although *ITGB1* was critical for directing the fate of cells during collagen-induced transdifferentiation, it was not necessary for initiating the transition from single cell to collective morphogenesis. Thus, it is not yet clear how cells sense the collagen architecture to initiate this process, but the response appears to be unique to stem-like cancer cells (MDA-MB-231 and HT-1080) as opposed to normal cells (HFF-1). Since, in our system, cells are embedded sparsely and undergo transcriptional reprogramming prior to cell division, the involvement of cell–cell interactions does not appear to have a role in transdifferentiation initiation. It is possible that cell interactions with the unique matrix architecture involve matrix sequestration of soluble factors and autocrine signaling. Indeed, TGFβ pathways were implicated by GO enrichment analysis (Fig. 2f). Alternatively, the initial confinement and rounded geometry of the cells enforced by the matrix may play a role. Several studies support a role for cellular geometry in numerous cellular processes including gene expression and differentiation[47–51], some of which is mediated by RhoA and cytoskeletal tension. However, confinement in Matrigel did not trigger the same process, indicating a unique requirement for cell–collagen interaction. Future work will address these questions.

ECM molecules, *COL4A1* and *LAMC2*, were also upregulated by CINP cells and have previously been implicated in driving migration and VM network formation in 2D culture[43, 44]. In our 3D collagen system, knockdown of either gene was not sufficient to block the VM-like phenotype (Supplementary Fig. 3). This suggests that regulation of in vitro cell network formation in a more physiological 3D culture context is distinct from regulation in a 2D culture context, which has implications for understanding molecular mechanisms. Given the significantly different requirements for cell movement in 3D ECM, such as matrix degradation and remodeling, our study highlights the importance of both the type of matrix and the dimensional context for studying physiological migration strategies. This echoes previous studies, which have shown that cell motility proteins function distinctly in a more physiologically relevant 3D context[20–22].

Interestingly, *SERPINE1*, a secreted protease inhibitor involved in coagulation and inflammation regulation, was upregulated by cancer cells as well as normal fibroblasts in response to confining collagen architectures. A recent study of cancer cell heterogeneity using mouse mammary carcinoma 4T1 cells and validated in human MDA-MB-231 breast cancer cells showed that cells which intrinsically expressed *SERPINE* family members were most efficient at spreading hematogenously, a characteristic that also correlated with their capacity to undergo VM in vivo[18]. Together, with our findings, this suggests that both cell-intrinsic and ECM factors may contribute to the emergence of VM. Interestingly, our

finding that fibroblasts and cancer cells both upregulate *SERPINE1* expression in confining collagen conditions hints at a potential supporting role for stromal cells in *SERPINE*-mediated VM metastasis[18].

The significant predictive value of our CINP gene signature in several tumor types may signify the physiological relevance of the ECM context and network-forming migration phenotype we created in vitro to a conserved mechanism of solid tumor metastasis. It is possible that gene expression analysis of additional cancer cell types induced into VM-like behavior by our 3D collagen system could help to further refine the conserved CINP gene module. This would facilitate prioritization of the genes for targeted functional studies to identify key regulators and potential therapeutic targets. In addition to regulators of the CINP, the conserved gene module also likely contains elements responsive to collagen but not directly involved.

Profiling additional cancer cell types and patient-derived tumor cells could also help to refine the gene module's prognostic value in the nine tumor types already identified or define additional cancer-specific versions of the CINP. Validation of the prognostic value of this gene module could help patients avoid the long-term side effects of aggressive radiation and chemotherapy if the likelihood of metastasis is very low. A recent meta-analysis of histological VM in over 3000 patients with various solid tumor types found that the visual presence of this cancer phenotype is specifically associated with poor prognosis[52]. Molecular detection of VM markers could provide a more quantitative measure.

## Methods

**Cell culture**. HT-1080 and HFF-1 were purchased from (ATCC, Manassas, VA) MDA-MB-231 cells were provided by Adam Engler (UCSD Bioengineering). All cell lines were cultured in high glucose Dulbecco's modified Eagle's medium supplemented with 10% (v/v) fetal bovine serum (FBS, Corning, Corning, NY) and 0.1% gentamicin (Gibco Thermofisher, Waltham, MA), and maintained at 37 °C and 5% $CO_2$ in a humidified environment during culture and imaging. The cells were passaged every 2–3 days. Cell culture under hypoxia was done on a humidified and temperature controlled environment at 1% $O_2$. Cells were tested for mycoplasma contamination using the Mycoalert kit (Lonza, Basel, Switzerland) before performing experiments.

**3D culture in collagen I matrix**. Cells embedded in 3D collagen matrices were prepared by mixing cells suspended in culture medium and 10× reconstitution buffer, 1:1 (v/v), with soluble rat tail type I collagen in acetic acid (Corning, Corning, NY) to achieve the desired final concentration[10, 20, 21]. A total of 1 M NaOH was used to normalize pH in a volume proportional to collagen required at each tested concentration (pH 7, 10–20 μl 1 M NaOH), and the mixture was placed in 48-well-culture plates and let polymerize at 37 °C. Final gel volumes were 200 μl.

**Cell tracking and motility analysis**. Cells were embedded in 3D collagen matrices in 48 well plates and left polymerize for 1 h in a standard tissue culture incubator and then 200 μl of complete growth medium were added on top of the gels. The gels were transferred to a microscope stage top incubator and cells were imaged at low magnification (×10) every 2 min for 48 h. Coordinates of the cell location at each time frame were determined by tracking single cells using image recognition software (Metamorph/Metavue, Molecular Devices, Sunnyvale, CA). Tracking data were processed using custom written python scripts based on previously published scripts[53] to calculate cell speed, invasion distances, and mean-squared displacements (MSDs). For cell motility analysis before and after division the time-lapse videos were scanned to identify dividing cells within the imaging period and the division point was identified as the frame at which a clear separation could be identified between daughter cells. The dividing cell was tracked up to the division point and one of the daughter cells (randomly chosen) was tracked from that point until the 48 h time point. For collective cell invasion distance, the 48 h time-lapse video was processed to obtain the maximum intensity projection (MIP), which highlights the tracks taken by the cells/groups of cells. Individual tracks distinguishable in the MIP were measured to obtain an equivalent invasion distance. All cell tracking data comes from three independent experiments performed on different days and with different cell passages.

**Persistence random walk model implementation**. To quantify the differences in the MSDs, we fitted the MSDs for each condition using the persistent random walk model (PRW model) as described in refs.[53, 54]. Briefly, the MSDs were calculated as in Eq. 1. The Eq. 2 describing the PWR was fitted using python's lmfit library for

each MSD. The persistent time (parameter P) was then extracted to calculate differences between groups as presented in Fig. 1a, b.

$$MSD(\tau) = \langle (x(t + \tau) - x(t))^2 + (y(t + \tau) - y(t))^2 \rangle, \quad (1)$$

}\eqno \eqno \text{(2)}

where, S is the cell speed and P is the persistence time and σ is a function of the error in the position of the cell as described in ref. [54].

**Collagen stiffness modification and measurement using shear rheology**. To modify the stiffness of collagen matrices without increasing density of material, we kept 2.5 mg ml$^{-1}$ gels at 20 °C for 30 min until they were fully polymerized. After the initial polymerization the gels were placed on a humidified tissue culture incubator at 37 °C for at least 1 hour extra before adding cell growth media on top. To measure the effect of polymerization temperature on the gel stiffness, we recreated the polymerization conditions for rheology testing (hybrid rheometer (DHR-2) from TA Instruments, New Castle, DE) using a cone and plate geometry with a sample volume of 0.6 ml. Shear storage modulus G′ was measured as reported before[10]. Briefly, we first performed a strain sweep was from 0.1 to 100% strain at a frequency of 1 rad s$^{-1}$ to determine the elastic region. Then a frequency sweep was performed at a strain within the linear region (0.8%) between 0.1 and 100 rad s$^{-1}$. Three independent replicates were performed for each condition tested.

**Collagen structure modification using polyethylene glycol**. To modify the structure of the collagen fibers within the gels without changing the final collagen concentration, Polyethylene glycol (PEG, MW = 8000, Sigma, St. Louis, MO) was solubilized in phosphate-buffered solution (PBS), filter sterilized. Solubilized PEG was then mixed into the cells, reconstitution buffer solution described above to produce a final PEG concentration of 10 mg ml$^{-1}$ in the collagen gel. The gels were allowed to polymerize in the same conditions as collagen only gels. Collagen structure modification was verified using confocal reflection microscopy.

**RNA Isolation and purification**. 3D collagen I gels were seeded in three independent experiments and harvested after 24 h of culture for RNA extraction and directly homogenized in Trizol reagent (Thermofisher, Waltham, MA). Total RNA was isolated following manufacturer's instructions. Isolated RNA was further purified using High Pure RNA Isolation Kit (ROCHE, Branford, CT). RNA integrity was verified using RNA Analysis ScreenTape (Agilent Technologies, La Jolla, CA) before sequencing.

**RNA sequencing and data analysis**. Biological triplicates of total RNA were prepared for sequencing using the TruSeq Stranded mRNA Sample Prep Kit (Illumina, San Diego, CA) and sequenced on the Illumina MiSeq platform at a depth of >25 million reads per sample. The read aligner Bowtie2 was used to build an index of the reference human genome hg19 UCSC and transcriptome. Paired-end reads were aligned to this index using Bowtie2[55] and streamed to eXpress[56] for transcript abundance quantification using command line "bowtie2 -a -p 10 -x /hg19 -1 reads_R1.fastq -2 reads_R2.fastq | express transcripts_hg19.fasta". For downstream analysis TPM was used as a measure of gene expression. A gene was considered detected if it had mean TPM >5.

**Gene ontology term overrepresentation analysis**. To assess the overrepresented GO terms the cytoscape app BiNGO[57] was used. Statistical test used was hypergeometric test, Benjamini–Hochberg false discovery rate (FDR) correction was used to account for multiple tests and the significance level was set at 0.05. For a given term, to assess the sensitivity of the enriched gene sets to the genes used in the analysis, we varied the threshold for including a gene as differentially upregulated from a fold change of 1.3 to a fold change of 1.9. The probability of a gene enriched with term is (# of genes in background with term)/(# of genes in background). The fold enrichment is the observed number of genes associated with term divided by the expected number of genes associated with term.

**Gene expression using qPCR**. For qPCR experiments, RNA was extracted as stated above and cDNA was synthesized using superscript iii first-strand synthesis system (Thermofisher, Waltham, MA). Relative mRNA levels were quantified using predesigned TaqMan gene expression assays (Thermofisher, Waltham, MA). Relative expression was calculated using the DCt method using GAPDH as reference gene. Assays used were: GAPDH (Hs02758991_g1), HIF1A (Hs00153153_m1), THBS1 (Hs00962908_m1), TGFBI (Hs00932747_m1), TPM1 (Hs04398572_m1), LAMC2 (Hs01043717_m1), and HMOX1 (Hs01110250_m1).

**Immunofluorescence and cell imaging**. For cell imaging after 7 days of culture to visualize VM structures collagen gels were fixed using two washes of 4% PFA for

30 min each at room temperature. F-actin was stained using AlexaFluor® 488 Phalloidin (Cell signaling technology, Danver, MA) and the nuclei were counterstained with DAPI. For immunofluorescence staining the gels were incubated with the primary antibody for 48–72 h. Anti-COL4A1 (1:200 dilution, NB120-6586, Novus Biologicals).

**Confocal reflection imaging and quantification**. Confocal reflection images were acquired using a Leica SP5 confocal microscope (Buffalo Grove, IL) equipped with a HCX APO L 20× 1.0 water immersion objective. The sample was excited at 488 nm and reflected light was collected without an emission filter. For the estimation of pore size we used modification of a previously reported digital imaging processing technique[10]. Briefly, the images were normalized to account for uneven illumination effects. Then a threshold was applied to generate a binary mask where pores were identified as the darkest areas of the image. Pore diameter was measured using NIS elements software (Nikon Instruments Inc., Melville, NY) measure objects tool.

**Gene suppression**. The lentiCRISPR v2 was a gift from Feng Zhang (Addgene plasmid #52961). We cloned small guide RNAs targeting the genes of interest into the lentiCRISPR v2 following Zhang's lab instructions. The sg_RNA sequences using were taken from the GECKO human library A[58]. Used sequences were: ITGB1 sg_RNA1 (5′-TGCTGTGTGTGTTTGCTCAAAC-3′), ITGB1 sg_RNA2 (5′-ATCTCCAGCAAAGTGAAACC-3′), EGFP sgRNA (5′-GGGCGAG-GAGCTGTTCACCG-3′). The lentiCRISPR v2 vectors with the cloned desired sgRNA were sequence verified and viral particles were generated by transfecting into lentiX293 T cells (Clonetech, Mountain View, CA. Cat #632180) along with packaging expressing plasmid (psPAX2, Addgene #12260) and envelope expressing plasmid (pMD2.G, Addgene #12259). Viral particles were collected at 48 h after transfection and they were purified by filtering through a 0.45 μm filter. Target cells were transduced with the viral particles in the presence of polybrene (Allele Biotechnology, San Diego, CA). After overnight incubation media was changed and cells were left 24–48 h in normal growth media and then changed to puromycin selection media (2.5 μg ml$^{-1}$ puromycin) for 7 days before experiments were performed. For shRNA-mediated gene knockdown, glycerol stocks of TRC2-pLKO.1-puro shRNA targeting LAMC2 (NM_005562.1-1019s1c1: CCGGGGCTCACCAA-GACTTACACATTCTCGAGAATGTGTAAGTCTTGGTGAGCTTTTTG), COL4A1 (NM_001845.3-3859s1c1:CCGGCCTGGGATTGATGGAGTTAAACTC-GAGTTTAACTCCATCAA TCCCAGGTTTTTG), and a non-targeting scramble sequence (SHC016:CCGGGCGCGATAGCGCTAATAATT TCTCGA-GAAATTATTAGCGCTATCGCGCTTTTT) were purchased from Sigma-Aldrich packaged in LentiX293T (Clonetech, Mountain View, CA. Cat #632180) along with packaging expressing plasmid as described above. Lentiviral particles were transduced into target cells and stably expressing cells were selected with puromycin (2 μg ml$^{-1}$) for at least 5 days before using.

**Western blotting**. Cells were grown to >90% confluency in 100 mm dishes. After washing 2X with PBS cells were collected into 100 μl of lysis buffer with 1× Halt protease inhibitor cocktail (Pierce IP lysis Buffer, Thermofisher, Waltham, MA) by thoroughly scraping the dish surface. Cell lysate was incubate in ice with constant shaking for 30 min and then centrifuged at 15,000×g for 20 for protein purification. Samples were loaded at 50 μg total protein concentration for SDS-PAGE. Membranes were probed with antibodies against ITGB1 (#4706 from Cell signaling technology, Danver, MA. 1:10,000 dilution) and α–Tubulin (TU-01 MA1-19162, Thermofisher, Waltham, MA. 1:30,000 dilution).

**Fluorescence-activated cell sorting**. Wild type MDA-MB-231 cells were grown in collagen I-coated tissue culture dished until 80% confluence. Cells were harvested using HyClone HyQtase (GE Healthcare Life Sciences, Marlborough, MA) and resuspended in FACS buffer (1% BSA, 0.5 mM EDTA in PBS). The cell suspension was then labeled using a monoclonal antibody against human CD29 (b1 integrin) conjugated to AlexaFluor 488. A cell suspension without added antibody was used as negative control. After labeling, the cells were analyzed within 1 h of detachment at the stem cell core of Sanford Consortium of Regenerative Medicine (La Jolla, CA) using a BD Influx cell sorter (BD, Franklin lakes, NJ). Cells were sorted based on fluorescence intensity into the top-expressing population (~15%, ITGB1 high) and bottom-expressing population (~13%, ITGB1 low). Sorted cells were replated into collagen-coated dishes and left to recover overnight. After recovery, the cells were embedded in 3D collagen gels as described above.

**Experimental data analysis and statistics**. All cell motility data were analyzed for statistical significance using the Scipy Python package. Additional experimental data were analyzed using prism Graphpad (San Diego, CA). Significance (p) was indicated within the figures using the following scale: *p < 0.05, **p < 0.01, ***p < 0.001. Additional relevant information is detailed in the figure captions.

**TCGA data reprocessing and survival analysis**. The TCGA raw data were downloaded from CGHub directly using gtdownload[59]. Corresponding clinical metadata were obtained from the TCGA data portal (https://tcga-data.nci.nih.gov/

docs/publications/tcga/). RNA-seq fastq files were realigned and quantified using sailfish v.0.7.6[60] with default parameters. Only primary tumors were considered in our analysis. In the analysis of breast invasive carcinoma, only the patients with reported histological staining for the three markers (Her2, ER, PR) could be associated with a molecular subtype. Patients for which any of the histological markers were not evaluated or were detected at an equivocal level were assigned to an "unknown" subtype. TCGA data for stage I, II, III, and IV breast cancer patients were analyzed by principal component analysis (PCA) with respect to the 70 CINP genes to construct gene expression meta-markers as previously described[61]. PCA-based score quantiles were mapped to CINP high and CINP low categories based on mean CINP gene expression levels. Because the CINP signature comprised only genes that were upregulated in the presence of the network phenotype, the overall mean expression of CINP genes was used to map PCA score to CINP signature activity level.

**METABRIC data retrieval and survival analysis**. We retrieved the clinical and microarray expression dataset from cBioPortal (http://www.cbioportal.org/study?id=brca_metabric). We were able to map 59 out of 70 CINP genes to METABRIC microarray data (missing genes: ZNF532, TRMT13, AMIGO2, KIN, NKX3-1, TANC2, TVP23C, SDHAP1, MTND2P28, GTF2IP4, H2BFS). Survival analysis was performed using the same method as described above for TCGA data. The Cox multiple regression uses CINP score, age, and three molecular subtype categories as covariates.

**TCGA pan cancer analysis**. Tumor types for which at least 100 patients had both expression and clinical metadata were analyzed to determine correlation between a CINP gene expression and 5-year survival. Only primary tumors were considered. Kaplan–Meier analysis was performed comparing the 30% of individuals with the lowest CINP expression score to the 30% with the highest score. The cox multiple regression uses age and CINP score as covariates. Both analyses use the Lifelines python library (https://lifelines.readthedocs.io/en/latest/). The log-rank test was used to determine significance of survival differences between groups.

**Human Protein Atlas data**. The online database Human Protein Atlas[46] was used to identify breast cancer tumor slices displaying hallmarks of the VM phenotype and subsequently assess protein expression of the genes associated with our in vitro network-forming phenotype. The tumor of patient ID 1910 was found to display linear chains of cancer cells lining interconnected matrix networks and had been stained for numerous other proteins of interest. Histological images shown in Fig. 5d can be found at www.proteinatlas.org by searching for the gene name in the breast cancer database and selecting patient ID 1910.

**Code availability**. Relevant scripts for the analysis of TCGA and METABRIC data are available at: https://github.com/brianyiktaktsui/Vascular_Mimicry.

**Data availability**. All sequencing data from this study has been deposited in the National Center for Biotechnology Information Gene Expression Omnibus (GEO) and is accessible through the GEO Series accession number GSE101209. All other relevant data are available within the article and supplementary files, or from the corresponding author upon request.

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

## Acknowledgements

We would like to acknowledge Kristen Jepsen and the IGM Genomics Core for assistance with RNA-seq experimentation, Christian Metallo for use of his hypoxia chamber, and Pedro Cabrales for use of his rheometer. The results shown here are in part based upon data generated by the TCGA Research Network: http://cancergenome.nih.gov/. S.I.F. and lab members are supported by a Burroughs Wellcome Fund Career Award at the Scientific Interface 1012027, NSF CAREER Award 1651855, and UCSD CTRI, FISP, and AIM pilot grants. H.C. and lab members are supported by the NIH grant DP5-OD017937 and UCSD FISP grant.

## Author contributions

D.O.V. conducted all cell and collagen matrix experiments and analyses. T.G., A.H., D.O.V. and C.L.C. performed PEG-collagen matrix experiments and analyses. D.O.V. and B.T. analyzed RNA sequencing data. B.T. performed TCGA and METABRIC analyses. C.L.C. conducted COL4A1 and LAMC2 knockdown studies. S.I.F. conceived and supervised the project. S.I.F. and H.C. co-designed RNA-sequencing experiments and co-supervised sequencing data analysis. All authors wrote or edited the manuscript.

## Additional information

**Competing interests:** The authors are inventors on a patent related to the prognositc gene signature reported herein.

