## [Peer Review File · Nature Communications]

Reviewers' comments:

Reviewer #1 (Remarks to the Author):

The manuscript by Ortiz and coworkers reports interesting findings generated by their 3D in vitro model system designed to probe the physical basis of cancer cell migration responses to collagen matrix organization - - using cultures of MDA-MB-231 breast cancer, HT-1080 fibrosarcoma, and HFF-1 human foreskin fibroblasts. The authors utilized soluble rat tail type I collagen, cell tracking and motility analyses, mathematical modeling, RNA sequencing, immunofluorescence and cell imaging, together with TCGA survival analysis - - to discover that dense, confining matrix architectures induced a migration behavior and transcriptional response leading to vasculogenic mimicry (VM) network formation predictive of poor clinical outcome. There are interesting aspects of this study that are worthy of further consideration for publication; however, additional data are needed to strengthen the impact of the study so that truly novel findings are reported rather than what could be perceived as confirmatory of previous reports. The following suggestions are offered to the authors in this context:

- 1) For the findings to have universal relevance with respect to other key studies performed using 3D matrices and VM as a functional metric, it will be important to understand if the observations are specific to rat tail type I collagen or related to other matrices such as Matrigel.
- 2) Also noteworthy, and in recognition of seminal studies by Werb and colleagues, do the confined matrices contain evidence of migratory/VM inducers by Western blot and/or IHC analyses?
- 3) 4 the study.
- 4) For the most part, the paper is well written with only minor spelling errors.

Reviewer #2 (Remarks to the Author):

The investigators have examined the impact of different 3D collagen type I structures onto the behavior and gene expression of two cancer cell lines and one fibroblast cell line. They find that a high density collagen structure promotes formation of a cancer cell network reminiscent of a vascular mimicry phenotype. Authors also find that the cancer cell lines starts to efficiently invade the 3D structure after one cell division in high density collagen, with a migration speed similar to that in low density collagen, and with a higher directional persistence. Further, sets of differentially expressed genes between high and low collagen density are identified and correlated to a vascular formation gene ontology and to disease outcome in cancer patients.

The differential effects of the different 3D-ECM structures are very interesting and of large potential interest, but at present, the molecular leads to what may cause the key phenotypes are limited to correlations with gene expression profiles. However, if functional molecular mechanistic data can be provided, in addition to added analysis and control experiments as specified below, I think this paper would be of high interest and priority.

1. The title claims that a vascular mimicry (VM) is induced "through a migratory and transcriptional response". However, at present, the vascular mimicry phenotype is only correlated to the migratory and transcriptional response, but a functional link is missing. Without intervention based experiments, it will remain unclear if the highlighted transcriptional and migratory response is actually responsible for the VM phenotype. To substantiate their main claim posted in the title, authors need to perform perturbations (such as by RNAi) of their DE genes (or a selection thereof) to identify molecular mechanisms responsible for the observed phenotypes of VM and migration. This would also provide functional molecular mechanistic information that would make this study significantly more interesting.
2. The gene ontology based enrichment analyses suffer from the low number of genes (70 and 35) included in the analysis. This makes the enrichment analysis extremely sensitive to random effects, since inclusion or exclusion of one single gene as being DE can dramatically alter the p-

value for a particular ontology enrichment. In combination with the inherent imperfections of gene ontology sets, this makes it important to use a very high stringency for enrichment of ontology gene sets and in addition to the p-values also to display and carefully judge the fold-enrichment and size of each ontology gene set.

3. For Figure 2 and 3; at least 3 biological repeats must be performed (and used for quantifications) to ensure reproducibility. In Figure 3, it is unclear how the quantifications were performed, since information on the number of experiments is lacking, as well as what the statistical analyses are based on. What is marked by the error bars?
4. What are the hazard ratios of patient survival? The hazard ratio is at least as important as the p-values to judge the significance of the effect on patient outcome.
5. Data is presented for five cancer types in which the authors found a correlation with patient outcome, which is a very interesting finding. Was such correlation found in all cancer types analyzed or were other cancer types uncorrelated to this gene signature?
6. Authors claim that the outcome prediction is independent of the breast cancer subtype (line 226). This statement is based on the distribution between different subtypes among the investigated patients. However, such conclusion cannot be made without directly comparing the correlation of the gene expression signature to survival outcome between the different breast cancer subtypes (KM analyses per subtype comparing the HRs). For example, the bars in Fig 4C-D may indicate an enrichment of low VM among HER2+ breast cancers. However, the present sample size would not allow such an analysis in a statistically adequate manner. To substantiate their claim of no difference between BC subtypes, the authors therefore need to analyze a much larger sample set - this should be straight forward, since there are multiple such datasets readily available (e.g. the Metabric dataset with approx. 2000 BC patients). The bar graphs of Fig 4C-D are not extremely useful at this stage and should be removed or moved to the supplementary information section.

Reviewer #3 (Remarks to the Author):

The manuscript by Ortiz et al. describes a refined 3D in vitro model that allows the authors to show a link between matrix architecture, migration phenotype (vascular mimicry) and gene expression program. The reviewer is not aware of any other biophysical work (very classical on cell migration in 3D) that goes to the investigation of the transcriptional response and analysis of human tumor datasets. In this regard, the authors should be complemented for their impressive work, which is a nice example on how biophysical approaches can be truly useful in cancer biology. Although I am not an expert (and therefore I cannot be very critical) in RNA sequencing data analysis, I really admire this pluridisciplinary effort between bio-engineering, biophysics and genomics. I am intimately convinced that these approaches are very promising. In consequence, to support this, I would recommend publication of the manuscript in Nat. Com. However, there is a couple of minor points that could be revised prior to publication. In order to demonstrate that matrix architecture is the main parameter that triggers vascular mimicry, the authors propose a series of experiments in which, they claim, all parameters are decoupled. From the first "control" experiments, they vary the polymerization temperature, which is interpreted as a change in matrix stiffness. Yet, clearly, both the stiffness and the pore size (which are generally related -see Yang et al. Biophys J 97:2051, 2009) are altered. Second, and this is THE point that I really don't understand in this manuscript, the authors claim that changing the matrix density and therefore the pore size enables them to evaluate the influence of hypoxia. As clearly written in the manuscript, the authors expect that matrices with small pore size will restrict the diffusion of oxygen molecules! This is true from a general point of view. But, when we image collagen matrix and read reports on their characterization (see again ref in Biophys J above), from 1 to 5 mg/ml and from 20 to 37°C, mesh size (Fig 4a, Yang et al BJ 2009) are between 3 and 15 µm. I cannot imagine that the diffusion of a molecule that is 100,000 times smaller than the pore size of a low density matrix, will be hindered in a high density matrix with a pore size that is "only" 10,000 times bigger than molecular size. The authors seem to detect a small effect on hypoxia. I would bet that the reason is different. Anyway, I would be curious of any kind of explanation.

Reviewer #4 (Remarks to the Author):

In this manuscript, the authors have presented the following:

- 1- They show that cancer cells are more migratory in high density type I rat tail tendon collagen.
- 2- The cancer cells become more vascular like (stretched), so called vascular mimicry (VM), in high collagen density.
- 3- They do RNA sequencing on cells in high vs. low density collagen.
- 4- Then they show that stiffness and hypoxia are not determinants of VM phenotype that they observed in vitro.
- 5- They eventually stratify patients survival based on the genes they found from RNA-Seq in step 3 above.

From my understanding:

Step 1 is done well.

In Step 2, they observe that cancer cells are more stretched and they relate that to VM, a rare (and maybe controversial) phenomenon in clinic. They didn't do more characterization to show that what they observe is actually VM as in patients. Not all stretched cells are VM. Cells could just align to collagen network, something they don't show in their images together with cells.

In Step 3, it's true that they show stiffness, hypoxia and pore size are not changing the VM phenotype, but the gene expression they see could still be the result of hypoxia, pore size, stiffness, and many other possible things that come after changing the density of collagen. Basically the logic isn't quite clear here. The genes they observe are NOT VM genes, they just correlate with the VM phenotype. Those 70 genes could be called high type I collagen density genes.

The work in step 4 including change of stiffness, change of pore size, hypoxia seem fine and is well presented. From all the options, they ruled out stiffness, hypoxia and pore size, so focal adhesion/integrin signaling seems to be the remaining option, but they don't show any data regarding integrins.

Step 5 seems problematic. They stratify patients based on those 70 genes, which are correlated to VM, but their relationship is not causal. They call these genes VM genes, and say that patients with those genes have different overall survival; a statement that is logically wrong.

Taken together, some of the conclusions seem overstated and may require more mechanistic studies. And again, the conclusions are really narrowly prescribed about the effects of density of only one of the ~300 ECM macromolecules.

We sincerely thank the reviewers for their constructive criticisms. We have addressed all concerns and believe that our manuscript is substantially improved by the additional data and analyses that we include. Please find our specific responses to each reviewer's comments below in blue text. We have also marked new text within the manuscript in blue for ease of reference.

Reviewer #1 (Remarks to the Author):

The manuscript by Ortiz and coworkers reports interesting findings generated by their 3D *in vitro* model system designed to probe the physical basis of cancer cell migration responses to collagen matrix organization - - using cultures of MDA-MB-231 breast cancer, HT-1080 fibrosarcoma, and HFF-1 human foreskin fibroblasts. The authors utilized soluble rat tail type I collagen, cell tracking and motility analyses, mathematical modeling, RNA sequencing, immunofluorescence and cell imaging, together with TCGA survival analysis - - to discover that dense, confining matrix architectures induced a migration behavior and transcriptional response leading to vasculogenic mimicry (VM) network formation predictive of poor clinical outcome. There are interesting aspects of this study that are worthy of further consideration for publication; however, additional data are needed to strengthen the impact of the study *so that truly novel findings are reported rather than what could be perceived as confirmatory of previous reports*. The following suggestions are offered to the authors in this context:

1) For the findings to have universal relevance with respect to other key studies performed using 3D matrices and VM as a functional metric, it will be important to understand if the observations are specific to rat tail type I collagen or related to other matrices such as Matrigel.

We now include experiments with Matrigel and more thoroughly compare our findings to prior VM studies. Previous studies used 2D Matrigel environments to study VM *in vitro*. Our new experiments show that 1) high density collagen uniquely induces the VM-like network forming phenotype in 3D culture conditions, and 2) genes implicated in regulating cell migration and the VM phenotype in 2D systems (namely LAMC2 and COL4A1) do not mediate the VM phenotype in 3D collagen. The following text has been added to the manuscript along with new data in Figure 1, H and I, and Supplementary Figure 3B:

“Previous pioneering studies have shown that several aggressive melanoma cell lines which produce VM *in vivo* also intrinsically form VM network structures when cultured on top of Matrigel in a 2D *in vitro* context^{16,17}. Recently, other aggressive tumor cell types have been shown to intrinsically form VM-like network structures on top of Matrigel¹⁸⁻²¹. Therefore, we sought to understand whether the network phenotype induced by a 3D collagen I environment was distinct from that induced by a 2D Matrigel environment. First, we tested to see if our cells formed network structures on top of Matrigel. Few cells aligned within the first 24hrs of culture, and nearly all cells aggregated after 72hrs (Fig. 1H). Next we embedded MDA-MB-231 cells inside of Matrigel, in 3D culture. In this context, cells did not form network structures but instead formed rough-edged, disorganized spheroids (Fig. 1I). Thus, high density collagen uniquely

induced the network forming phenotype in a more physiologically relevant 3D context. Given the significantly different requirements for cell movement in 3D ECM, such as matrix degradation and remodeling, this finding highlights the importance of both the type of matrix and the dimensional context for studying physiological migration strategies. Likewise, previous studies have shown that cell motility proteins function distinctly in a more physiologically relevant 3D context²²⁻²⁴ .”

“Finally, we asked if upregulated genes in our transcriptional module that have previously been implicated as drivers of VM *in vitro* were functionally active in our network forming phenotype. LAMC2 (Ln-5, gamma 2 chain) was previously found to be upregulated in aggressive melanoma cells that intrinsically display the VM phenotype compared to less aggressive melanoma cells that don’t display VM. Moreover, it was implicated as a driver of VM network formation, since the cleavage of this secreted matrix molecule by MMP-2 and MT1-MMP produces pro-migratory fragments. In 2D culture of aggressive melanoma cells on top of collagen I, the inhibition of LAMC2 cleavage blocked VM network formation⁴⁴. Using shRNA to knock down LAMC2, we found that LAMC2 KD cells maintain their ability to form network structures in 3D high density collagen (Supplementary Fig. 3B). COL4A1 is another matrix molecule upregulated by cells undergoing the network phenotype (Fig. 1G and Fig. 2G) and previously implicated in driving migration⁴⁵. COL4A1 KD also did not inhibit the ability of cells to form network structures in 3D high density collagen (Supplementary Fig. 3B). This suggests that regulation of *in vitro* VM network formation in a more physiological 3D culture context is distinct from regulation in a 2D culture context, which has implications for understanding the molecular mechanisms driving the phenotype.”

2) Also noteworthy, and in recognition of seminal studies by Werb and colleagues, do the confined matrices contain evidence of migratory/VM inducers by Western blot and/or IHC analyses?

It is unclear what inducer molecules the reviewer is referring to. LAMC2 (Ln-5, gamma 2 chain) has been implicated as a driver of VM, as the cleavage of this secreted matrix molecule by MMP-2 and MT1-MMP produces pro-migratory fragments. LAMC2 is also found to be upregulated in aggressive melanoma cells that intrinsically display the VM phenotype compared to less aggressive melanoma cells that don’t display VM. In 2D culture of aggressive melanoma cells on top of collagen I, the inhibition of LAMC2 cleavage blocked VM network formation (Seftor REB, et.al. Cancer Research 61:17, 2001). We address the reviewer’s comment in this context with new experiments wherein LAMC2 is knocked down in our cells, which upregulate this gene as they are induced to switch into a VM phenotype by the collagen matrix architecture. We show that KD of LAMC2 does not inhibit VM network formation in our 3D culture context (new Supplementary Figure 3B). This suggests that regulation of *in vitro* VM network formation in a 2D culture context is distinct from its regulation in a more physiological 3D culture context, which has implications for understanding the molecular mechanisms of VM. We also knock down COL4A1, which is implicated in inducing migration, and show that it does not hinder *in vitro* 3D VM network formation. See above response to Reviewer 1 Comment 1 for the text that was added to the

manuscript concerning these new results. In recognition of seminal studies by Werb and colleagues, we also now cite their work describing the regulation of mammary epithelial morphogenesis by matrix remodeling (new reference 33).

3) Equally important, does evidence exist in patient tumor sections regarding the gene/protein signature associated with VM revealed in this report? This is a critical point that would tie together seminal studies in the field with the current one. Critics will claim you can generate different gene profiles related to different matrices, and the translational relevance will reside in the patient tumor samples associated in phenotype with the respective cell lines used in the study.

We now show IHC staining for several of the most upregulated genes in the module we identified in breast cancer patient primary tumor sections where VM is observed (new Figure 5D). THBS1, EDN1, and JAG1 antibodies stain VM cancer cells with strong intensity. This new data indicates that several genes in our panel are indeed expressed in patient tumors by cancer cells undergoing phenotypic VM.

This new data further strengthens the other clinical evidence we presented, wherein we used RNA-seq data from primary patient tumors and showed that patient survival is predicted with high statistical confidence in multiple tumor types by our 70 gene set. Together, these findings suggest substantial translational relevance of the gene signature for prognosticating patient outcomes, which could help guide treatment strategies.

New text has been added to the manuscript as follows:

“Finally, we sought to determine whether the *in vitro* network forming phenotype and associated transcriptional signature were related to the *in vivo* VM phenotype. Using the Human Protein Atlas (www.proteinatlas.org)⁴⁸, we first identified breast cancer tumor slices displaying hallmarks of the VM phenotype, namely linear chains of cells lining glycogen-rich matrix networks that conduct blood flow but do not stain positively for CD31¹⁶. The tumor of patient 1910 displayed linear chains of cancer cells lining interconnected matrix networks (Fig. 5D). An immunohistochemical stain for GYPA showed red blood cells flowing through the matrix-networks in tumor tissue but highly concentrated in vessel-like structures in healthy tissue. A stain against CD31 showed that there were no endothelial cells lining the matrix networks in the tumor tissues. Although a PAS stain was not available in the protein atlas database, which would determine whether the matrix networks were positive for glycogen, a stain against glycogen synthase (GSK3A) was available and showed that the chains of cancer cells significantly expressed this enzyme. The network forming cell phenotype combined with IHC evidence are consistent with the previously described histopathology of VM¹⁶. Next, we asked whether highly upregulated genes in our 70 gene CINP module were evident at the protein level in this clinical sample of VM. Stains for THBS1, JAG1, and EDN1 were available in the protein atlas database for the same tumor and showed significant expression of all three genes from our CINP transcriptional module in the VM tumor tissue but little stain in healthy tissues. Taken

together, this data suggests that clinical VM and our *in vitro* collagen induced network phenotype share many commonalities.”

4) For the most part, the paper is well written with only minor spelling errors. We identified spelling errors and corrected them.

Reviewer #2 (Remarks to the Author):

The investigators have examined the impact of different 3D collagen type I structures onto the behavior and gene expression of two cancer cell lines and one fibroblast cell line. They find that a high density collagen structure promotes formation of a cancer cell network reminiscent of a vascular mimicry phenotype. Authors also find that the cancer cell lines starts to efficiently invade the 3D structure after one cell division in high density collagen, with a migration speed similar to that in low density collagen, and with a higher directional persistence. Further, sets of differentially expressed genes between high and low collagen density are identified and correlated to a vascular formation gene ontology and to disease outcome in cancer patients.

The differential effects of the different 3D-ECM structures are very interesting and of large potential interest, but at present, the molecular leads to what may cause the key phenotypes are limited to correlations with gene expression profiles. However, if functional molecular mechanistic data can be provided, in addition to added analysis and control experiments as specified below, I think this paper would be of high interest and priority.

1. The title claims that a vascular mimicry (VM) is induced “through a migratory and transcriptional response”. However, at present, the vascular mimicry phenotype is only correlated to the migratory and transcriptional response, but a functional link is missing. Without intervention based experiments, it will remain unclear if the highlighted transcriptional and migratory response is actually responsible for the VM phenotype. To substantiate their main claim posted in the title, authors need to perform perturbations (such as by RNAi) of their DE genes (or a selection thereof) to identify molecular mechanisms responsible for the observed phenotypes of VM and migration. This would also provide functional molecular mechanistic information that would make this study significantly more interesting.

We now include new experiments and data functionally linking the gene expression profile with the observed phenotype. Since integrin $\beta 1$ (ITGB1) was upregulated in the transcriptional response to confining conditions (Fig. 2B) and mediates adhesion and migration on collagen, we asked whether the persistent migration phenotype observed in confining matrix conditions that leads to network formation was dependent on the upregulation of ITGB1. Through new experiments, we found that both the network forming phenotype and the associated gene expression was dependent on ITGB1 upregulation. The following text has been added to the manuscript along with a new Figure 4.

“The short, more isotropic arrangement of fibers associated with both the high density collagen and low density PEG crowded collagen conditions could act on cells through local cell-matrix interactions transduced by integrin signaling. Integrin $\beta 1$ (ITGB1) is a canonical receptor for collagen I and a central node in ECM signal transduction. Moreover, ITGB1 was upregulated by both cancer cell types in response to confining matrix conditions (Fig. 2B). Thus, we next asked whether the network forming phenotype observed in confining matrix conditions was mediated by ITGB1. CRISPR-Cas9 technology was used to silence ITGB1 expression with single guide RNAs (sgRNAs), and constructs expressing sgRNAs targeting eGFP were used as controls (Fig. 4A). Silenced and control cells were embedded separately and sparsely in low and high density collagen matrices. Cells were monitored by timelapse microscopy for early migration behavior then imaged again after one week. In low density collagen, ITGB1 silenced cells maintained a similar level of migration capability to WT cells in low density matrices, but used an amoeboid blebbing migration phenotype instead of a mesenchymal migration phenotype (Fig. 4B). In high density conditions, ITGB1 silenced cells migrated faster than WT cells, but were significantly less persistent and did not invade (Fig. 4C). Surprisingly, after one week ITGB1 silenced cells in high density collagen formed spheroid structures instead of cell networks, whereas control cells exhibited the same behavior as the wild type in both collagen conditions (Fig. 4D). Retrospective analysis of WT MDA-MB-231 cells in high density collagen revealed that a small fraction spontaneously formed spheroid structures (Fig. 4E). These findings suggest that either basal expression level or upregulation of ITGB1 dictates the network forming phenotype. To distinguish between these two possibilities, we next sorted the parental WT population based on basal ITGB1 expression level and then embedded high and low expressing cells separately in confining high density collagen matrices (Fig. 4F). We observed no appreciable differences in the percentage of networks versus spheroids formed by the sorted populations after one week. However, ITGB1 low cells proliferated less and displayed fewer total number of network or spheroid structures (Fig. 4G, and data not shown) even though the initial seeding density was the same (Supplementary Fig. 3A). Together, these results suggest that ITGB1 upregulation drives the persistent migration phenotype leading to network formation, but also that a transition from single cell behavior to multicellular structure formation is triggered by the confining conditions independently of ITGB1. More broadly, our findings also show that collagen fiber architecture dictates the role ITGB1 plays in migration. In one architectural context, ITGB1 facilitates a switch from mesenchymal to amoeboid migration and in another architectural context it mediates the shape of structures formed by collective migration behaviors.

To further explore the link between the upregulated transcriptional module and the network forming phenotype, we asked whether ITGB1 silenced spheroid forming cells showed different gene expression patterns than WT network forming cells. To assess this,

we conducted qRT-PCR analysis of a subset of the 70-gene panel in the two cell phenotypes. Upregulation of several key genes were maintained in the spheroid forming cells, while other genes were no longer upregulated (Fig. 4H). These results suggest that ITGB1 mediates a feedback loop regulating some aspects of the transcriptional module associated with the network forming phenotype.”

Also see response to Reviewer 1, Comments 1 and 2, where we discuss new data involving KD of the DE gene LAMC2, which was previously implicated in providing pro-migratory signals to VM cancer cells in a 2D culture environment, and show that in our 3D system LAMC2 is not required for *in vitro* VM network formation. We also KD the DE gene COL4A1, which was previously implicated in driving invasive migration of cancer cells, and show that it is not required for network formation. Together, these results suggest that ITGB1 upregulation is necessary for VM, but other upregulated genes are not drivers of this phenotype. Nonetheless, the set of 70 upregulated genes represent a response signature to a confining collagen matrix and feedback through ITGB1 upregulation that predicts patient outcomes. Text additions to the manuscript are noted above.

2. The gene ontology based enrichment analyses suffer from the low number of genes (70 and 35) included in the analysis. This makes the enrichment analysis extremely sensitive to random effects, since inclusion or exclusion of one single gene as being DE can dramatically alter the p-value for a particular ontology enrichment. In combination with the inherent imperfections of gene ontology sets, this makes it important to use a very high stringency for enrichment of ontology gene sets and in addition to the p-values also to display and carefully judge the fold-enrichment and size of each ontology gene set.

We thank the reviewer for this suggestion. We now report the fold enrichment and gene set sizes as well as the p-value. To assess the sensitivity of the enriched gene sets to the genes used in the analysis, we varied the threshold for including a gene as differentially upregulated from a fold change of 1.3 to a fold change of 1.9. The terms highlighted in the manuscript, including “blood vessel development”, “regulation of cell migration” from the 70 genes, and “cell differentiation”, “regulation of smooth muscle cell migration” from the 35 genes, were robust to the expression threshold change (Supplementary Fig. 2D).

New text has been added to the manuscript as follows:

“Importantly, changes in the threshold for differential expression did not significantly alter the primary gene ontology categories identified (Supplementary Fig. 2D).”

3. For Figure 2 and 3; at least 3 biological repeats must be performed (and used for quantifications) to ensure reproducibility. Three biological replicates were performed and used for quantification. A statement clarifying this point has been added to the figure captions and methods. In Figure 3, it is unclear how the quantifications were performed, since information on the number of experiments is lacking, as well as what the statistical analyses are based on. What is marked by the error bars? We apologize for the oversight. Details of our quantification and statistical analysis are now provided in the figure caption and in the methods section.

4. What are the hazard ratios of patient survival? The hazard ratio is at least as important as the p-values to judge the significance of the effect on patient outcome.

We now report hazard ratios for each breast cancer patient stage analysis using the METABRIC database, since this database includes sufficient data for such an analysis. Hazard ratios are estimated by fitting a cox proportional hazards model with molecular subtypes, age, and the CINP metagene score as covariates (Fig. 5, A and B, and Supplementary Fig. 4, C and D). Stage I, II, III, and IV breast cancer data in TCGA does not include a sufficient number of events/deaths to enable fitting a cox proportional hazards model with these same 5 covariates. For additional cancer types analyzed using TCGA, age and CINP score were used as covariates (Fig. 5C and Supplementary Fig. 4E). Importantly, this new analysis and associated hazard ratios are consistent with our initial findings.

New text has been added to the manuscript as follows in blue type:

“Finally, we sought to determine if the collagen induced network phenotype (CINP) triggered by our 3D system was clinically relevant. To test this, we first asked whether the 70 common-to-cancer genes associated with the CINP could predict cancer patient prognosis. We anticipated that if this gene signature was indicative of a more metastatic cancer cell migration phenotype, its expression would correlate with poor patient outcomes. Since late stage tumors are already characterized by migration of tumor cells to distant lymph nodes or organs, we hypothesized that a gene signature associated with metastatic migration would correlate with prognosis in early (Stage I & II) but not late (Stage III & IV) stage tumors. Using the cancer genome atlas (TCGA), we first analyzed data for breast cancer patients with respect to the expression of the 70 gene signature. An expression metagene was constructed using the loadings of the first principal component (CINP PC1) of a 195 Stage I patient by 70 gene matrix (Supplementary Fig. 4A, also see methods). Then a survival analysis was conducted, comparing patients with the highest (top 30%) and lowest (bottom 30%) expression metagene scores by log rank test. The cumulative survival rate of these two groups differed significantly (log rank $p=0.049$); however, there was insufficient data to power a hazard ratio (HR) calculation (Fig. 5A). Analysis using the more data-rich METABRIC microarray database of breast cancer patients showed similar results for Stage I, confirming the prognostic value of the gene set (log-rank $p=0.037$, HR=1.40, Cox $p=0.002$, Fig. 5A). Applying the same analysis to Stage II breast cancer patients revealed that the CINP metagene was associated with a marginally significant difference in 5-year survival by TCGA analysis but not by METABRIC analysis (Supplementary Fig. 4C). One caveat to this analysis is that data for 11 of the genes in our 70 gene panel were not available in the METABRIC dataset. The CINP metagene also did not separate patients with better prognosis in late stage tumors (Supplementary Fig. 4D). These results indicate that the CINP gene module could have clinical predictive power in the early stages of breast cancer.”

5. Data is presented for five cancer types in which the authors found a correlation with patient outcome, which is a very interesting finding. Was such correlation found in all cancer types analyzed or were other cancer types uncorrelated to this gene signature?

In our original analysis, we included LGG (n=510), CESC(n=304), LUAD (n=524) and PAAD (n=180) because their sample sizes were reasonably large. To further quantify the necessary minimum sample size to yield robust finding in a given disease, we performed a power analysis. This analysis suggested that detecting a hazard ratio of 1.5 with 80% power would require nearly 200 samples, ideally evenly split between CINP high and CINP low categories. Thus we hesitate to draw conclusions about tumor types with fewer than ~200 samples in the Stage I and II categories. Nonetheless, we now include a table of all cancers in TCGA and report the number of patients, number of deaths, hazard ratios, and Cox model p-values. For these cancer types, all stages and subtypes were used in the analysis, and age and CINP score were used as covariates. In addition to the cancers we previously reported, we observe a significant effect of CINP in MESO, ACC, BLCA, and KICH, but had previously excluded them due to their small sample size. Moreover, a list of cancers for which the gene module was not predictive are included in Supplementary Figure 4E. Overall our results are suggestive that our CINP signature may have broad implications for many tumor types.

New text has been added to the manuscript as follows:

“Next, we screened the predictive value of the gene module in additional cancer types in TCGA independently of stage or subtype using age and CINP score as covariates. The CINP gene module was a significant predictor of survival in lung adenocarcinoma (LUAD), lower grade glioma (LGG), cervical squamous cell carcinoma and endocervical adenocarcinoma (CESC), pancreatic adenocarcinoma (PAAD), mesothelioma (MESO), adrenocortical carcinoma (ACC), bladder urothelial carcinoma (BLCA), and kidney chromophobe carcinoma (KICH) (Fig. 5C), but was not a significant predictor in several other tumor types found in TCGA (Supplementary Fig. 4E). The significant predictive value of our CINP gene signature in several tumor types may signify the physiological relevance of the ECM context and network forming migration phenotype we created *in vitro* to a conserved mechanism of solid tumor metastasis.”

6. Authors claim that the outcome prediction is independent of the breast cancer subtype (line 226). This statement is based on the distribution between different subtypes among the investigated patients. However, such conclusion cannot be made without directly comparing the correlation of the gene expression signature to survival outcome between the different breast cancer subtypes (KM analyses per subtype comparing the HRs). For example, the bars in Fig 4C-D may indicate an enrichment of low VM among HER2+ breast cancers. However, the present sample size would not allow such an analysis in a statistically adequate manner. To substantiate their claim of no difference between BC subtypes, the authors therefore need to analyze a much larger sample set - this should be straight forward, since there are multiple such datasets readily available (e.g. the Metabric dataset with approx. 2000 BC patients). The bar graphs of Fig 4C-D are not extremely

useful at this stage and should be removed or moved to the supplementary information section.

We thank the reviewer for this very helpful suggestion. We have now investigated the association between the CINP metagene score and breast cancer molecular subtypes in the METABRIC dataset. We were able to replicate the association of a high CINP score with poor patient outcome in stage I METABRIC patients. Fitting a cox model to the Stage I METABRIC data with subtypes, age, and CINP score as covariates, CINP score had the highest hazard ratio (hazard ratio=1.42). We also found that the CINP score was more informative in triple-negative and luminal A patients compared with other patients in METABRIC (Fig. 5B). This analysis suggests that the CINP score is not entirely independent of molecular subtype in breast cancer but is still an informative covariate that should be further explored.

New text has been added to the manuscript as follows:

“Importantly, further analysis of Stage I patients by molecular subtype⁴⁶ revealed that the CINP metagene provided significant prognostic value for Luminal A and Triple Negative breast cancer patients (Fig. 5B). This analysis suggests that the CINP score is not entirely independent of molecular subtype in breast cancer but is still an informative covariate.”

Reviewer #3 (Remarks to the Author):

The manuscript by Ortiz et al. describes a refined 3D in vitro model that allows the authors to show a link between matrix architecture, migration phenotype (vascular mimicry) and gene expression program. The reviewer is not aware of any other biophysical work (very classical on cell migration in 3D) that goes to the investigation of the transcriptional response and analysis of human tumor datasets. In this regard, the authors should be complemented for their impressive work, which is a nice example on how biophysical approaches can be truly useful in cancer biology. Although I am not an expert (and therefore I cannot be very critical) in RNA sequencing data analysis, I really admire this pluridisciplinary effort between bio-engineering, biophysics and genomics. I am intimately convinced that these approaches are very promising. In consequence, to support this, I would recommend publication of the manuscript in Nat. Com. However, there is a couple of minor points that could be revised prior to publication. In order to demonstrate that matrix architecture is the main parameter that triggers vascular mimicry, the authors propose a series of experiments in which, they claim, all parameters are decoupled. From the first "control" experiments, they vary the polymerization temperature, which is interpreted as a change in matrix stiffness. Yet, clearly, both the stiffness and the pore size (which are generally related -see Yang et al. Biophys J 97:2051, 2009) are altered.

We did not mean to claim that all parameters are entirely decoupled. This is nearly impossible to accomplish within a 3D protein network. Nonetheless, we can conclude that cells cultured within the stiffened matrix do not undergo the transformation. To clarify, we now add further details describing the pore size and fiber architecture of the stiffened matrix. The new text reads as follows:

“By lowering the polymerization temperature from 37°C to 20°C, polymerization slowed, allowing fibers to form more organized and reinforced fiber structures with larger pores (data not shown).”

Second, and this is THE point that I really don't understand in this manuscript, the authors claim that changing the matrix density and therefore the pore size enables them to evaluate the influence of hypoxia. To directly evaluate the influence of hypoxia, we culture cells in a low oxygen incubator. As clearly written in the manuscript, the authors expect that matrices with small pore size will restrict the diffusion of oxygen molecules! This is true from a general point of view. But, when we image collagen matrix and read reports on their characterization (see again ref in Biophys J above), from 1 to 5 mg/ml and from 20 to 37°C, mesh size (Fig 4a, Yang et al BJ 2009) are between 3 and 15 μm . I cannot imagine that the diffusion of a molecule that is 100,000 times smaller than the pore size of a low density matrix, will be hindered in a high density matrix with a pore size that is "only" 10,000 times bigger than molecular size. The authors seem to detect a small effect on hypoxia. I would bet that the reason is different. Anyway, I would be curious of any kind of explanation. The consumption of oxygen by cells in 3D scaffolds can outpace the diffusion of oxygen from the air-liquid interface through the liquid and gel layers. A statement and references clarifying this point has been added to the manuscript as follows:

“One way in which smaller pore sizes could influence cell behavior is by restricting the diffusion of molecules to and from the cells³⁶. More specifically, the imbalance between oxygen diffusion to cells and oxygen consumption by cells in 3D matrices has been shown to promote hypoxic conditions in some cases³⁷.”

Reviewer #4 (Remarks to the Author):

In this manuscript, the authors have presented the following:

- 1- They show that cancer cells are more migratory in high density type I rat tail tendon collagen.
- 2- The cancer cells become more vascular like (stretched), so called vascular mimicry (VM), in high collagen density.
- 3- They do RNA sequencing on cells in high vs. low density collagen.
- 4- Then they show that stiffness and hypoxia are not determinants of VM phenotype that they observed in vitro.
- 5- They eventually stratify patients survival based on the genes they found from RNA-Seq in step 3 above.

From my understanding:

Step 1 is done well.

In Step 2, they observe that cancer cells are more stretched and they relate that to VM, a rare (and maybe controversial) phenomenon in clinic. They didn't do more characterization to show that what they observe is actually VM as in patients. Not all stretched cells are VM. Cells could just align to collagen network, something they don't show in their images together with cells.

We appreciate the reviewer's concern and now include clinical evidence showing that several of our upregulated genes are highly expressed in breast tumor slices displaying VM. Please see response to Reviewer 1, Comment 3 for the new data and text that has been added to the manuscript to address this point.

We have also added a reflection confocal image of the collagen surrounding the network forming cells in our 3D collagen system (new Supplementary Figure 1E), which shows that the cell network does not simply align to the collagen fibers. The following text has been added to the manuscript for clarification of this point:

“Interestingly, these network structures do not appear to be caused by cells aligning along collagen fibers (Supplementary Fig. 1E).”

In Step 3, it's true that they show stiffness, hypoxia and pore size are not changing the VM phenotype, but the gene expression they see could still be the result of hypoxia, pore size, stiffness, and many other possible things that come after changing the density of collagen. Basically the logic isn't quite clear here. The genes they observe are NOT VM genes, they just correlate with the VM phenotype. Those 70 genes could be called high type I collagen density genes.

We thank the reviewer for their constructive criticism. Through new experiments wherein the upregulated gene integrin $\beta 1$ (ITGB1) is knocked down, we show that indeed a portion of the gene signature is related to the network phenotype and is mediated by feedback through ITGB1 upregulation, while another portion appears to be more generally related to the collagen condition. Please see response above to Reviewer 2, Comment 1 for the exact text and data additions to the manuscript.

We have also attempted to clarify the way in which we refer to the gene set and phenotype to better indicate their relationships. We now say that the gene set is associated with the “Collagen Induced Network Phenotype” or CINP.

The work in step 4 including change of stiffness, change of pore size, hypoxia seem fine and is well presented. From all the options, they ruled our stiffness, hypoxia and pore size, so focal adhesion/integrin signaling seems to be the remaining option, but they don't show any data regarding integrins.

We now include new experiments wherein ITGB1 is silenced as well as cell populations sorted based on ITGB1 protein levels. Please see response to Reviewer 2, Comment 1 above for the exact text and data that has been added to the manuscript.

Step 5 seems problematic. They stratify patients based on those 70 genes, which are correlated to VM, but their relationship is not causal. They call these genes VM genes, and say that patients with those genes have different overall survival; a statement that is logically wrong.

We now show that several genes in the upregulated gene set are expressed highly at the protein level in patient tumors displaying VM. We have also clarified our wording concerning this gene set, now referring to it as “genes associated with the collagen induced network phenotype (CINP)”. For full explanation of the new clinical validation data and text that has been added to the manuscript, please see above response to Reviewer 1, Comment 3.

Taken together, some of the conclusions seem overstated and may require more mechanistic studies. And again, the conclusions are really narrowly prescribed about the effects of density of only one of the ~300 ECM macromolecules.

As detailed above, we have added significant new data to further support our conclusions, but we have also attempted to tone down language throughout the manuscript that may have been interpreted as overstating.

We also now include experiments with other matrix molecules, namely Matrigel, and show that our system is unique in inducing VM-like network formation in breast cancer cells. Please see response to Reviewer 1, Comment 1 for details and the exact text and data that has been added to the manuscript.

Reviewers' comments:

Reviewer #1 (Remarks to the Author):

The revised manuscript by Ortiz and coworkers has attempted to respond to the myriad comments from four Reviewers, which is a challenging task. However, in many respects, select additional data and associated interpretation of previous seminal studies are somewhat biased and incorrect (in certain cases). Studying vasculogenic mimicry (VM) primarily at the in vitro level can lead to interesting biological observations that require validation in situ, preferably in patient tissues. However, the in vitro approach has its limitations when trying to compare data across a plethora of studies using various matrices with different concentrations of ECM components, various observation windows of time, diverse cell lines, etc. In addition, there are remarkable discrepancies that exist in the quality and consistency of commercial products used in experimental matrices. Therefore, one cannot be too dogmatic about a direct driver of VM because there are many drivers across various tumor types.

The following points should be addressed if this manuscript proceeds to the next level of review:

- In the Abstract, "vascluogenesis" should be changed to "vasculogenesis."
- The upregulation of B1 integrin in tumor cell progression is not novel.
- Particularly troubling is the misinterpretation of "previous pioneering studies" showing "that several aggressive melanoma cell lines which produce VM in vivo intrinsically form VM network structures when cultured on top of Matrigel in a 2D in vitro context." A careful review of these studies indicates that the model and observations were based on the microscopic demonstration of melanoma cells plated on and in Matrigel or Type I collagen gels; histological cross-sections of these thick 3D matrices revealed cells within the matrices and VM initially as vascular cords, followed by tubular formation, and luminization -- capable of conductance of perfusion dyes (observations made over 14 days in culture). These VM data were confirmed by TEM and SEM. The manuscript under review does not contain this type of detail nor scientific rigor.
- With respect to the MDA-MB-231 cells, the authors may wish to re-examine the data reported by Liu and colleagues in Oncogene showing VM on and in Matrigel. This is reference #20 in the manuscript, and it too is misrepresented as an example of tumor cells forming VM-like network structures (only) on top of Matrigel with no mention of VM in Matrigel, as clearly shown in this paper.
- Lastly, the authors might be interested in a meta-analysis of tumor VM in over 3,000 patients – specifically associated with poor prognosis (by Cao and colleagues).
- The primary novel finding of this paper is related to the identification of a transcriptional response common to multiple cancer cell types. It is not earth-shattering, but it is new information. Unfortunately, this is overshadowed by the lack of credibility associated with some biased statements relevant to previous work – seemingly done to reinforce a myopic point of view. There is no reason to reinvent the literature on this topic.

Reviewer #2 (Remarks to the Author):

NCOMMS-16-28802A-Z

The authors have made significant efforts to improve this study, efforts that for the most part satisfy my previous concerns. However, in my view there are still a few issues that must be corrected to make this manuscript acceptable for publication.

1. The authors have added experiments where the ITGB1 gene was deleted that link the integrin b1 gene to the observed phenotype and altered gene expression. Also, an impressive effort was made to address the ITGB1 upregulation as such (in addition to complete gene deletion), which makes an important distinction. These experiments clearly add value by defining a specific upregulated gene that determines the functional outcome.

At the same time, other genes upregulated in the expression profile were depleted, but with no functional outcome. In their rebuttal letter, the authors therefore correctly state: "Together, these results suggest that ITGB1 upregulation is necessary for VM but other upregulated genes are not drivers of this phenotype".

In contrast, the title still states that the transcriptional response mediated by ITGB1 is responsible for the phenotypes. ("3D collagen architecture induces vascular mimicry in cancer cells through a conserved migratory and transcriptional response mediated by integrin $\beta 1$ "). The link between the transcriptional response (other than ITGB1) and the functional outcomes needs to be removed.

2. The testing of different threshold of DE genes for testing of ontology enrichment is a valuable addition. However, the authors should also add threshold for the minimum number of enriched genes in each ontology gene set. This becomes most clear in the cases where a single gene in an ontology gene set appears as DE; it is apparent that this cannot imply enrichment of a specific ontology, but only represent the gene itself.

Given also the imperfection of ontologies, were many ontology gene sets include genes that are quite distant to the indicated function, conclusions of enrichment based on only a few genes should be avoided. Therefore, the "regulation of smooth muscle cell migration" enrichment based on only 3 genes is questionable and should be filtered out.

3. The authors have added deepened analysis of the implication of their CINP score for different breast cancer molecular subtypes; these results are much more convincing than originally because of the improved statistical methodology and sample size in the analysis. The authors find that the CINP score is significant for two of the BC subtypes, but not for others. This makes the authors to conclude that "the CINP score is not entirely independent of molecular subtype in breast cancer". This appears to this reviewer as an innovative way to express that the value of the CINP score may be limited to certain BC subtypes. I recommend the quoted sentence to be removed.

Reviewer #3 (Remarks to the Author):

My previous report was already mostly positive about this work.

Upon reading of the response to other reviewers (more focused on fields in which I am not expert) it seems to me that the authors did significantly improve their manuscript by performing and analyzing numerous new experiments.

Concerning my major concern (pore size of the gel that may trigger hypoxia), even though the authors did not directly address the point (how a difference by several orders of magnitude between oxygen size and pore size in all cases can affect hypoxia), they refer to a published work, which I have to take even though no mechanistic reason is provided.

In consequence, I think that this work is worthy of being published in Nature Comm.

Reviewer #4 commented for the editors only and was satisfied with the revised manuscript.

We sincerely thank the reviewers for their time, in depth review of our manuscript, and constructive comments. We believe we have addressed all concerns, and that our manuscript is significantly improved by these additions. New response text is shown in green here and in the manuscript. Response text from the previous revision is still shown in blue.

Reviewer #1 (Remarks to the Author):

The revised manuscript by Ortiz and coworkers has attempted to respond to the myriad comments from four Reviewers, which is a challenging task. However, in many respects, select additional data and associated interpretation of previous seminal studies are somewhat biased and incorrect (in certain cases). Studying vasculogenic mimicry (VM) primarily at the *in vitro* level can lead to interesting biological observations that require validation *in situ*, preferably in patient tissues.

We appreciate the need for continued investigation and plan to do so. However, we note that we have already included evidence in patient tissues and in patient tumor sequencing datasets.

However, the *in vitro* approach has its limitations when trying to compare data across a plethora of studies using various matrices with different concentrations of ECM components, various observation windows of time, diverse cell lines, etc. In addition, there are remarkable discrepancies that exist in the quality and consistency of commercial products used in experimental matrices. Therefore, one cannot be too dogmatic about a direct driver of VM because there are many drivers across various tumor types.

We appreciate the limitations of *in vitro* model systems and understand the reviewer's concern. These limitations are the primary reason we sought to compare our *in vitro* findings to clinical patient data (tissues and sequencing). We have also made efforts to present our findings conservatively by referring to our observation as "VM-like". Moreover, we have concluded, "this data suggests that clinical VM and our *in vitro* collagen induced network phenotype share many commonalities."

Further, we have now changed our title to be more conservative:

"3D collagen architecture induces a conserved migratory and transcriptional response linked to vasculogenic mimicry and mediated by integrin $\beta 1$ upregulation"

We agree with what the reviewer mentions about the variability of ECM reagents and approaches as well as the lack of experimental details such as timelines, concentrations, matrix architectures, etc. These factors do make it challenging to compare and contrast findings. However, we have presented in this manuscript

integrative multidisciplinary evidence coming from several different approaches that support the role of the collagen in triggering a VM-like phenomenon.

To address the reviewer's concerns, we have added the following statement to the manuscript text:

"It is important to note that variations exist in the consistency of commercial ECM products...."

The following points should be addressed if this manuscript proceeds to the next level of review:

- In the Abstract, "vascluogenesis" should be changed to "vasculogenesis." We have corrected this spelling error.
- The upregulation of B1 integrin in tumor cell progression is not novel.

We did not claim that ITGB1 upregulation in tumor progression is novel. Our findings do provide a novel link between ITGB1 upregulation and a persistent migration phenotype, cell network formation, and accompanying transcriptional changes. Moreover, we show that the role of ITGB1 in modulating cell migration is matrix context dependent. In one architectural context, ITGB1 facilitates a switch from mesenchymal to amoeboid migration and in another architectural context it mediates the shape of structures formed by collective migration behaviors. To address the reviewer's concern and more clearly acknowledge prior work on the role of ITGB1 in tumor progression, we have added the following statement to the manuscript to:

"Integrin α 1 (ITGB1) is a canonical receptor for collagen I and a central node in ECM signal transduction. Prior studies have identified ITGB1 as a critical mediator of breast cancer progression in mouse and *in vitro* models⁴¹."

- Particularly troubling is the misinterpretation of "previous pioneering studies" showing "that several aggressive melanoma cell lines which produce VM *in vivo* intrinsically form VM network structures when cultured on top of Matrigel in a 2D *in vitro* context." A careful review of these studies indicates that the model and observations were based on the microscopic demonstration of melanoma cells plated on and in Matrigel or Type I collagen gels; histological cross-sections of these thick 3D matrices revealed cells within the matrices and VM initially as vascular cords, followed by tubular formation, and luminization -- capable of conductance of perfusion dyes (observations made over 14 days in culture). These VM data were confirmed by TEM and SEM. The manuscript under review does not contain this type of detail nor scientific rigor.

The study the reviewer is referring to, "Vascular Channel Formation by Human Melanoma Cells *In Vivo* and *In Vitro*: Vasculogenic Mimicry", is to our knowledge the

first description of the vascular mimicry phenomenon. It is, as the reviewer points out, a very in depth, detailed, and rigorous study. In the methods section of this article, Maniotis et. al. described their 3D culture system as:

“Three-Dimensional Cultures

Twelve microliters of Matrigel or Type I collagen (Collaborative Biomedical) were dropped onto glass coverslips and allowed to polymerize for 1 hour at 37°C. Tumor cell lines, normal uveal melanocytes, or endothelial cells were then seeded **on top of the gels** at high density and allowed to incubate.”

This is what we define as a planar, 2D, monolayer culture system on top of an ECM matrix. We define a 3D culture system as one where the cells are embedded fully, in contact on all sides with the ECM matrix, and far from the influence of the stiff planar coverslip bottom of the dish. We define a 2.5D culture system as pseudo 3D, in which the cells are embedded in the matrix but in contact with the coverslip bottom or sides of the dish. We have previously demonstrated the importance of these distinctions, as cell behavior and protein localization are differentially regulated in each context (please see references 19-21).

We would also like to point out that the data presented in this manuscript adds additional support for the importance of these naming conventions:

“Interestingly, cells that were in contact with the coverslip and not fully embedded in the high density condition did not undergo the same migration transition upon division (Supplementary Fig. 1, A and B).”

We have included the following clarifying statement to address the reviewer’s concerns:

“It is important to note that variations exist in the consistency of commercial ECM products as well as the terminology used to describe 3D culture. Here, we define 3D culture strictly as a condition where cells are fully embedded, in contact with ECM on all sides, and located a sufficient distance away from the coverslip bottom and sides of the culture dish to avoid their influence. We define 2.5D culture as a pseudo 3D culture where cells are embedded in the ECM but in contact with coverslip. Our previous studies have demonstrated the importance of these distinctions, as cell behavior and protein localization are differentially regulated in each context¹⁹⁻²¹.”

- With respect to the MDA-MB-231 cells, the authors may wish to re-examine the data reported by Liu and colleagues in Oncogene showing VM on and in Matrigel. This is reference #20 in the manuscript, and it too is misrepresented as an example of tumor cells forming VM-like network structures (only) on top of Matrigel with no mention of VM in Matrigel, as clearly shown in this paper.

We believe the difference in our interpretation of this paper is again due to a difference in naming conventions. The paper by Lui and colleagues shows a single bright field image of MDAs in what is described as a 3D culture, but elongated cells are in a single focal plane, indicating that they are located on the coverslip bottom. This is a pseudo 3D condition that we previously denoted as 2.5D, because cells in this condition are functionally distinct from cells that are fully embedded in 3D (see references 19-21). Additionally, Lui et. al. present no further characterization of the cells in this condition beyond the single image. So it is unclear whether the cells were truly in a VM-like state. To clarify this for the reader, we have added a more in depth description of our definition of 3D culture, which is supported by our previous publications (see response above). In addition, we have changed our description of the Liu et al experiments for consistency:

“Recently, other aggressive tumor cell types have been shown to intrinsically form VM-like network structures on top of Matrigel or in 2.5D Matrigel culture¹⁵⁻¹⁸.”

- Lastly, the authors might be interested in a meta-analysis of tumor VM in over 3,000 patients – specifically associated with poor prognosis (by Cao and colleagues).

We thank the reviewer for pointing out this interesting study. This study reinforces the importance of VM as a biomarker for solid tumor metastatic potential and as a possible therapeutic target. We have added a comment and properly cited this study in our manuscript:

“A recent meta-analysis of VM in over 3,000 patients with various solid tumor types found that the presence of this cancer phenotype is specifically associated with poor prognosis¹⁴.”

- The primary novel finding of this paper is related to the identification of a transcriptional response common to multiple cancer cell types. It is not earth-shattering, but it is new information. Unfortunately, this is overshadowed by the lack of credibility associated with some biased statements relevant to previous work – seemingly done to reinforce a myopic point of view. There is no reason to reinvent the literature on this topic.

We hope we have clarified for the reviewer our intention to be as technically accurate as possible in light of the important differences that have been demonstrated among the culture conditions we are comparing.

Reviewer #2 (Remarks to the Author):

NCOMMS-16-28802A-Z

The authors have made significant efforts to improve this study, efforts that for the most part satisfy my previous concerns. However, in my view there are still a few issues that must be corrected to make this manuscript acceptable for publication.

1. The authors have added experiments where the ITGB1 gene was deleted that link the integrin b1 gene to the observed phenotype and altered gene expression. Also, an impressive effort was made to address the ITGB1 upregulation as such (in addition to complete gene deletion), which makes an important distinction. These experiments clearly add value by defining a specific upregulated gene that determines the functional outcome.

At the same time, other genes upregulated in the expression profile were depleted, but with no functional outcome. In their rebuttal letter, the authors therefore correctly state: "Together, these results suggest that ITGB1 upregulation is necessary for VM but other upregulated genes are not drivers of this phenotype". In contrast, the title still states that the transcriptional response mediated by ITGB1 is responsible for the phenotypes. ("3D collagen architecture induces vascular mimicry in cancer cells through a conserved migratory and transcriptional response mediated by integrin β 1"). The link between the transcriptional response (other than ITGB1) and the functional outcomes needs to be removed.

We thank the reviewer for this comment and have changed our title as follows:

"3D collagen architecture induces a conserved migratory and transcriptional response linked to vasculogenic mimicry and mediated by integrin β 1 upregulation"

We used the word "linked" here to represent our evidence of the transcriptional response gene products in histological tumor sections and because the gene set as a whole shows prognostic value in multiple tumor types.

2. The testing of different threshold of DE genes for testing of ontology enrichment is a valuable addition. However, the authors should also add threshold for the minimum number of enriched genes in each ontology gene set. This becomes most clear in the cases where a single gene in an ontology gene set appears as DE; it is apparent that this cannot imply enrichment of a specific ontology, but only represent the gene itself.

Given also the imperfection of ontologies, were many ontology gene sets include genes that are quite distant to the indicated function, conclusions of enrichment based on only a few genes should be avoided. Therefore, the "regulation of smooth muscle cell migration" enrichment based on only 3 genes is questionable and should be filtered out.

We understand the reviewer's concern that the categories with few genes may not be relevant. As the reviewer rightfully mentions, conclusions based on a few genes may or may not be relevant depending on how closely the genes are involved in the indicated function. However, this makes it difficult to identify a single threshold that can be uniformly applied. To address this concern and offer the reader a more complete picture of the data, we now include the number of genes per category in

the GO enrichment analysis figures (Figures 2F and 2H) and also provide the gene identities in each category as a new supplementary table.

We have also removed our reference to the smooth muscle cell migration enrichment category in the manuscript text, since it only contained 3 genes.

We also note that the categories containing low numbers of genes are only present in the 35 gene list which represents genes upregulated by all 3 cell lines tested (Figure 2H) and not in the 70 gene list associated with the VM phenotype (Figure 2F).

Finally, to clarify the limitations of GO enrichment for the readers, we have added the following statement:

“It is important to take into account the inherent flaws associated with GO enrichment analysis. For example, some categories showing enrichment in the 35 genes common to all cell lines contain very few genes and may not represent real enrichment. However, this limitation is not observed in the top enriched categories in the 70 genes common to cancer cells. The genes associated with each enrichment category are shown in Supplementary Tables 1 and 2.”

3. The authors have added deepened analysis of the implication of their CINP score for different breast cancer molecular subtypes; these results are much more convincing than originally because of the improved statistical methodology and sample size in the analysis. The authors find that the CINP score is significant for two of the BC subtypes, but not for others. This makes the authors to conclude that “the CINP score is not entirely independent of molecular subtype in breast cancer”. This appears to this reviewer as an innovative way to express that the value of the CINP score may be limited to certain BC subtypes. I recommend the quoted sentence to be removed.

We have removed the quoted sentence.

Reviewer #3 (Remarks to the Author):

My previous report was already mostly positive about this work. Upon reading of the response to other reviewers (more focused on fields in which I am not expert) it seems to me that the authors did significantly improve their manuscript by performing and analyzing numerous new experiments. Concerning my major concern (pore size of the gel that may trigger hypoxia), even though the authors did not directly address the point (how a difference by several orders of magnitude between oxygen size and pore size in all cases can affect hypoxia), they refer to a published work, which I have to take even though no

mechanistic reason is provided.

In consequence, I think that this work is worthy of being published in Nature Comm.

Reviewer #4 commented for the editors only and was satisfied with the revised manuscript.

REVIEWERS' COMMENTS:

Reviewer #1 (Remarks to the Author):

The revised manuscript by Fraley and colleagues contains several improvements over the previous version. However, two important issues remain to be addressed to ensure credibility and avoid perpetuation of incomplete information/misinterpretation of an impressive body of literature related to VM:

1) Contrary to the authors' interpretation of my comment that "Particularly troubling is the misinterpretation of previous pioneering studies...", I was not referring to the original 1999 VM report - - rather, the quintessential studies following the original paper performed with various matrices (including 3D matrices) of sufficient thickness to perform histological cross-sections of paraffin-embedded samples, demonstrating VM networks within the matrices. The 1999 VM paper gave birth to a new field where there were many questions to address in subsequent studies, and myriad tools, including laser microdissection, TEM, and confocal microscopy were used to examine the tumor cells on top of and within various matrices.

2) Based on the fact that tumor cell lines express varying degrees of COL4A1 and LAMC2, it is critical, for the sake of credibility and neutrality, that the authors qualify their observations by stating "in the MDA-MB-231 and HT1080 cell lines examined in this study." Unless the authors intend to study the effects of down-regulating COL4A1 and LAMC2 in other cell lines, it is unfair to compare apples with oranges, especially when the observations are different in other cell lines.

Reviewer #2 (Remarks to the Author):

The authors have adequately addressed all my concerns.

We want to thank the reviewers again for their interest in our work and for the time they have dedicated to providing constructive reviews. We have addressed all remaining specific concerns, as detailed below.

REVIEWERS' COMMENTS:

Reviewer #1 (Remarks to the Author):

The revised manuscript by Fraley and colleagues contains several improvements over the previous version. However, two important issues remain to be addressed to ensure credibility and avoid perpetuation of incomplete information/misinterpretation of an impressive body of literature related to VM:

1) Contrary to the authors' interpretation of my comment that "Particularly troubling is the misinterpretation of previous pioneering studies...", I was not referring to the original 1999 VM report - - rather, the quintessential studies following the original paper performed with various matrices (including 3D matrices) of sufficient thickness to perform histological cross-sections of paraffin-embedded samples, demonstrating VM networks within the matrices. The 1999 VM paper gave birth to a new field where there were many questions to address in subsequent studies, and myriad tools, including laser microdissection, TEM, and confocal microscopy were used to examine the tumor cells on top of and within various matrices.

We were forced to interpret the reviewer's comment because the reviewer provided no specific details. Unfortunately, the reviewer has again not specified the papers to which they are referring, making it impossible for us to specifically address their concerns. This generalized format of criticism is not constructive. We have reviewed and cited the relevant literature spanning from the original report of VM in 1999 to 2016. Although several of the papers we cite (in addition to the seminal 1999 paper) use the terminology "3D culture" in their figure captions, a careful review of their methods sections reveal that cells were actually plated "on top" of 3D matrices, which does not meet the field's current definition of 3D culture. We discuss in detail in our manuscript the current widely accepted definitions of 3D culture so that no confusion arises. As we stated previously, this definition is supported by the fact that protein localization and function are differentially regulated in 3D and 2D and 2.5D contexts (references 19-21 in the manuscript).

2) Based on the fact that tumor cell lines express varying degrees of COL4A1 and LAMC2, it is critical, for the sake of credibility and neutrality, that the authors qualify their observations by stating "in the MDA-MB-231 and HT1080 cell lines examined in this study." Unless the authors intend to study the effects of down-regulating COL4A1 and LAMC2 in other cell lines, it is unfair to compare apples with oranges, especially when the observations are different in other cell lines.

We understand the reviewer's concern and have added the requested qualifying statement as a tracked change in the manuscript word document.

Reviewer #2 (Remarks to the Author):

The authors have adequately addressed all my concerns.

We sincerely thank the reviewer for their constructive review and believe that their comments helped to improve our paper.